PLOS · Biology

# Reward expectation yields distinct effects on sensory processing and decision making in the human brain

Ankita Sengupta*, Devarajan Sridharan *

Centre for Neuroscience, Indian Institute of Science, Bangalore, India

* ankita@iisc.ac.in (AS); sridhar@iisc.ac.in (DS)

## Abstract

Reward expectation robustly guides both attention and decisions. Yet, whether common or distinct mechanisms mediate each of these processes remains unknown. Previous studies have often conflated the effect of reward expectation on sensory processing and decision-making because locations selected for sensory prioritization (sensitivity effects) were also prioritized for decisions (criterion effects). Here, we identify distinct forms of reward expectation that separably control spatial attention and decisional biases in human cortex. Sensitivity and criterion were independently modulated when expected rewards varied across locations ("space-specific") or choices ("choice-specific"), respectively. Only sensitivity, not criterion, modulations reflected a limited, conserved attentional resource. Established neural and physiological signatures of attention, including gain modulation of event-related potentials, alpha-band power lateralization, and eye-movement biases, were elicited only by space-specific reward modulation. By contrast, neural correlates of decisional biases, including pre-stimulus alpha power suppression, selectively accompanied choice-specific reward modulation. Attention-related neural markers predicted sensitivity modulation by space-specific reward expectation but not criterion modulation by choice-specific reward expectation, indicating their distinct underlying mechanisms. Our findings uncover fundamentally dissociable behavioral and neural underpinnings of reward expectation effects on sensory and decisional selection, with critical implications for understanding how reward, attention, and choice are linked in the human brain.

## Author summary

Expecting a reward can shape behavior in diverse ways. For example, monkeys may focus their attention on tree branches containing the most plentiful fruit to identify ripe ones based on visual features, such as color or texture. However, they must also choose to act quickly and pick this fruit to thwart their

provided the original author and source are credited.

**Data availability statement:** The data generated during this study and custom code for reproducing each figure panel in this paper are available at: https://doi.org/10.6084/m9.figshare.25966015.

**Funding:** This research was supported by a Department of Biotechnology-Wellcome Trust India Alliance Intermediate fellowship (IA/I/15/2/502089, https://www.indiaalliance.org/, a Department of Science and Technology Swarna Jayanti Fellowship (SB/SJF/2021-22/02, https://dst.gov.in/scientific-programmes/scientific-engineering-research/human-resource-development-and-nurturing-young-talent-swarnajayanti-fellowships-scheme), a Pratiksha Trust Intramural grant (KVCH/22/2047, https://indiabioscience.org/), a Gore Subraya Bhat Chair Associate Professorship in Digital Health (https://odaa.iisc.ac.in/gore-subraya-bhat-chair-associate-professor-in-digital-health/), an India-Trento Program for Advanced Research grant (INT/ITALY/ITPAR-IV/COG/2018/G, https://www.indiascienceandtechnology.gov.in/bilateral-cooperations/department-science-technology-dst/italy) (all to DS), and an Institute of Eminence (IoE) grant, funded by the University Grants Commission (to IISc). The funders had no role in study design, data collection and analysis, decision to publish, or preparation of the manuscript.

**Competing interests:** The authors have declared that no competing interests exist.

**Abbreviations :** ACC, anterior cingulate cortex; ERP, event related potential; fMRI, functional magnetic resonance imaging; GSR, galvanic skin response; LIP, lateral intraparietal sulcus; OFC, orbitofrontal cortex; PFC, prefrontal cortex; SDT, signal detection theory.

conspecifics. In other words, reward expectation can influence not only how attention is engaged but also how choices are made. Yet, these two effects have been frequently conflated in laboratory tasks. Here, with a task that decouples reward expectation's effects on attention from those on decision-making, we uncover their distinct neural correlates. Our results show how reward shapes attention and biases choices independently in the human brain.

## Introduction

In everyday life, our actions are governed by both tangible and intangible incentives, like food, shelter, money, or social acceptance [1,2]. The value associated with such incentives is dynamic and often changes across space and time [2]. Even the expectation of future reward incentives can powerfully shape behavior: such reward expectation provides a strong cue for guiding our attention as well as how we make decisions. Reward incentives can improve performance by engaging top–down attentional selection to targets at high-reward locations [3–5] or by biasing our choices toward those associated with higher rewards [6–8]. A significant body of work has investigated behavioral and neural links between reward expectation, attention, and decision-making [9–12]. Yet, whether these cognitive phenomena are mediated by shared or distinct neural mechanisms is controversial [13].

Many previous studies have investigated how reward expectation modulates attention and decision-making for high-value targets. Higher reward expectation – in terms of either magnitude or probability – biases attention and decisions in favor of these targets, as evidenced by faster reaction times [3,14] and increased detection rates [4,15]. Moreover, other studies have shown that such behavioral modulations can be achieved with both rewards and penalties. For example, in a covert attention task with three types of reward incentive trials (win, lose, neutral), participants showed faster reaction times for rewards (win) and penalties (lose) than in the neither (neutral) incentive trials [3]. Reward expectation also robustly modulates brain activity [3,16–18], including in the prefrontal cortex (PFC) [10,18] and in the lateral intraparietal sulcus (LIP) [19,20].

In perceptual decision-making tasks, commonly employed in the lab, improved performance for the rewarding target can arise from enhanced perceptual processing for the target, improved (or biased) decision-making for the target, or a combination of both [4,11,12]. These effects can be quantified using the framework of signal detection theory (SDT) [21]: the index of sensitivity ($d'$) – the signal-to-noise ratio of the target's perceptual representation – quantifies sensory prioritization, whereas the choice criterion ($c$) – a threshold value for the target's evidence for downstream choices – quantifies decisional prioritization [4,11,12]. Here, we seek to distinguish the behavioral and neural consequences of reward-driven selection on sensory (sensitivity) from decisional (criterion) effects. Moreover, we test if either or both effects of reward-driven selection produce neural signatures conventionally associated with those of visual spatial attention [22–25].

These questions remain largely unanswered because task designs in the literature often fail to distinguish reward expectation effects of prioritized sensory processing from those of prioritized decision-making [20,26], or even motoric preparation for choices that reflect the outcome of the decision [19,27]. For example, a seminal study varied the rewards assigned to two alternative target locations independently during a dynamic foraging task and found that neural activity in the monkey lateral intraparietal sulcus (LIP) tracked the relative reward values between the locations [27]. Another study showed that LIP neurons tracked the motivational salience – associated with either rewards or penalties – rather than relative reward values alone [19]. However, neither study decoupled the perceptual consequences – in terms of improved sensitivity at the location of higher reward or salience – from the decisional consequences – in terms of criterion effects – driven by differences in reward across locations. Moreover, LIP is involved in both attentional allocation as well as in motor planning and execution [28]. As a result, the precise cognitive correlates of neural activity changes in the LIP remain unclear.

A notable exception is a related pair of recent studies [11,12] that investigated mechanisms of reward expectation using SDT in monkey visual and prefrontal cortex. During an endogenous change detection task, these authors manipulated either the relative reward between two locations or the absolute rewards between choices at the two locations, and observed either only sensitivity or only criterion modulations, respectively [11,12]. The authors found that overlapping neuronal populations in prefrontal cortex (PFC) encoded reward-induced changes in both sensitivity and criterion, but fluctuations in V4 activity were correlated only with changes in sensitivity, not with criterion shifts.

On the other hand, a recent study, also in monkeys, dissociated neural mechanisms for sensory versus decisional prioritization by employing a cued attention task with an anti-saccade response [29]. In this study, subjects (monkeys) performed a four-alternative orientation change detection task with a highly informative probabilistic cue and reported an orientation change at any location by making a saccade toward the location opposite to it (anti-saccade). Their results demonstrated a clear, double dissociation such that sensitivity and criterion effects occurred at the anti-saccade target and probabilistically cued locations, respectively. Importantly, and in contrast to earlier studies [11,12], they found that V4 firing rates were modulated by criterion, but not sensitivity changes, at the attentionally cued spatial location. The authors reconciled their results with these earlier findings [11,12] by hypothesizing that not all criterion-related manipulations reflect the operation of spatial attention [29]. Yet, this hypothesis remains to be directly tested with data.

Here, we test this hypothesis with a novel behavioral task design to distinguish the effects of reward expectation on attentional (sensitivity) versus decisional (criterion) selection and to identify their respective neural underpinnings. In our task, the reward contingency remained constant across trials at one ("fixed") location, whereas it switched between one of two alternatives randomly after short runs of trials, at the other ("variable") location (see "Results", for details). Participants implicitly and dynamically learned the reward contingency on the variable side based on trial-level feedback about their response accuracies. Moreover, they had to keenly monitor their performance on this side to identify when a reward contingency change occurred. With this task, we manipulated reward expectations in one of two distinct ways.

In one session, we varied the relative reward across stimulus locations ("space-specific") by maintaining a constant reward for correct responses at the fixed location, and a higher or lower overall reward in the variable location across short trial runs (Figs 1C–1E and S1B). We hypothesized that this manipulation would result in the prioritization of particular locations for sensory processing and spatial attention. Specifically, we expected a dynamic modulation of sensitivity at the variable reward location, with a higher sensitivity at this location when it possessed greater expected reward. We did not expect any effects on criterion because correct responses (hits and correct rejections) were rewarded equally at each respective location (described in detail next) (Figs 1D and S1B).

In another session, we varied the relative reward across response choices at the variable location ("choice-specific"). In this case, the overall reward expectation was equal across the two locations, but the two correct response choices (hits and correct rejections) were unequally rewarded on the variable side, with hits being rewarded more than correct rejections, and vice versa, in alternating, short trial runs. We hypothesized that this manipulation would produce dynamic

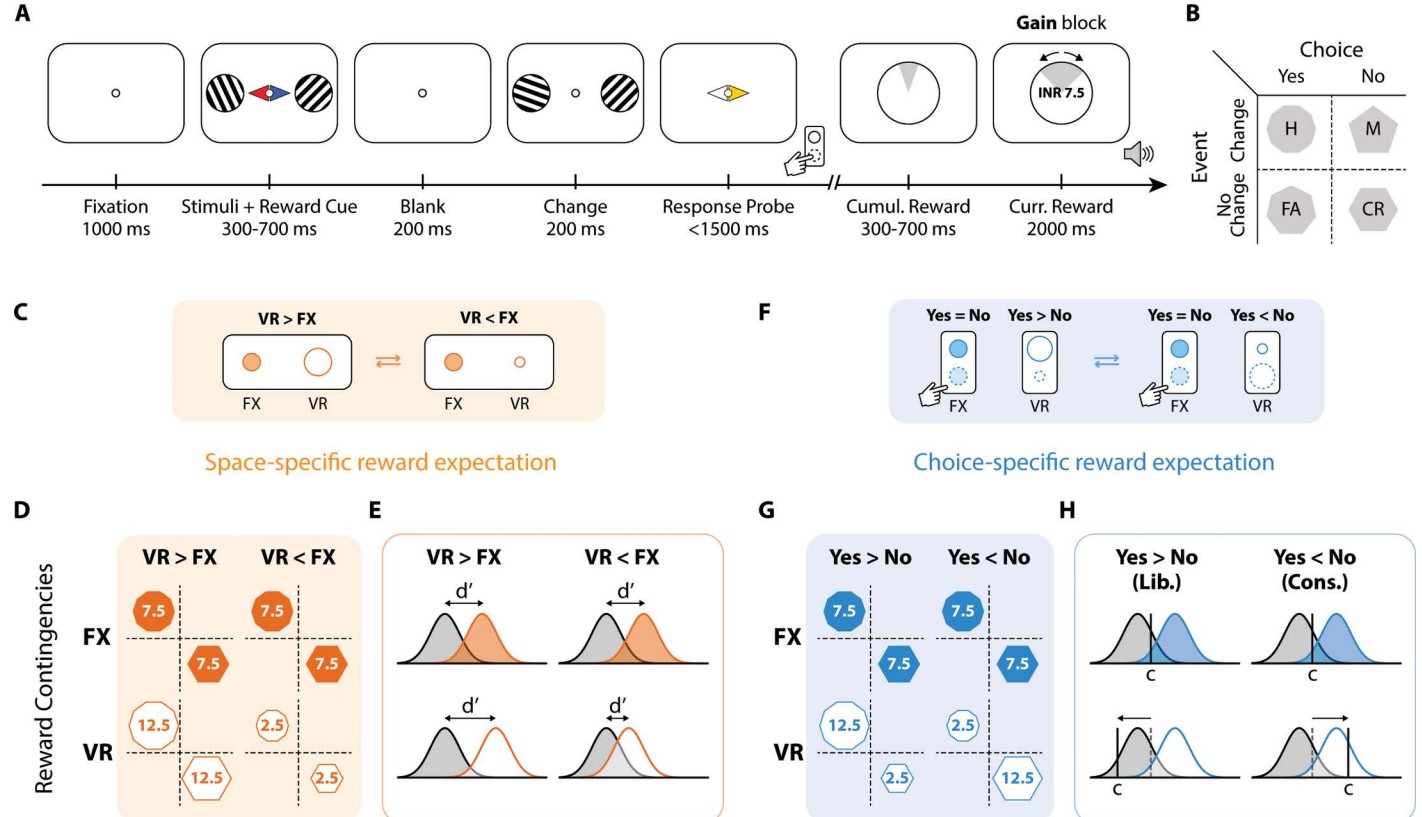

**Fig 1. Dissociating space-specific from choice-specific reward expectation. A.** Schematic of reward-cued change detection task. Participants were cued (central colored arrowheads) about expected reward contingencies (fixed or variable, see panels **D, G**) for correct responses on each side, and reported the occurrence of an orientation change on the probed side (yellow arrowhead). Audio-visual feedback indicated the current trial's reward and the cumulative reward accrued in the trial block (see text for details). **B.** Stimulus-response contingency table. Rows: event types. Columns: choice types. Shapes: Response types; octagon: hits/H, pentagon: misses/M, heptagon: false alarms/FA, hexagon: correct rejections/CR. **C–E.** Space-specific reward expectation session. **C.** Schematic of reward contingencies, in "mini-blocks", for the space-specific reward expectation session ("gain" blocks only; see text for details). **D.** Rows: reward contingencies on the fixed (FX; top row, filled polygons) and variable (VR; bottom row, open polygons) reward sides for the space-specific reward expectation session. Columns: reward contingency conditions, "VR > FX" (left) or "VR < FX" (right). Each cell: contingency table, with the same conventions as in panel **B**. Numbers within polygons indicate the reward (INR) for the respective (correct) response type. Blank represents no reward for the incorrect response types. **E.** One-dimensional signal detection theory (SDT) models showing hypothesized modulations of sensitivity ($d'$) for space-specific reward expectation. Rows and columns: same conventions as in panel **C**. Gaussians: decision variable distributions at the respective location and reward contingency. Gray and orange: noise and signal distributions (FX, filled; VR, open). Horizontal arrows: perceptual sensitivity ($d'$). **F–H.** Choice-specific reward expectation session. **F.** Schematic of reward contingencies, in "mini-blocks", for the choice-specific reward expectation session ("gain" blocks only; see text for details). **G.** Same as in panel **D** but for the choice-specific reward expectation session. Columns: reward contingency conditions; greater expected reward for correct Yes responses ("Yes > No"/"Liberal"), or vice versa ("Yes < No"/"Conservative"). **H.** Same as in panel **E**, but for choice-specific reward expectation. Row and column conventions are as in panel **G**. Solid vertical lines: choice criterion **(C)**. Arrows: Hypothesized criterion change for each reward contingency. **(G–H)**. Other conventions are the same as in panels **D and E**, respectively.

modulations of criterion at the variable, but not at the fixed location, without concomitant changes in sensitivity at either location. Importantly, we tested whether the location prioritized for decision-making – the variable reward location requiring dynamic criterion shifts – would exhibit signatures of selective attention, as compared to the fixed location. In other words, in both sessions, participants needed to regularly monitor changes in the reward contingency on the variable side. Thus, the variable reward side was always more important, even if not more valuable, than the fixed reward side in terms of guiding selection for sensory processing (space-specific) or for decision-making (choice-specific).

PLOS Biology

Additionally, we investigated neural correlates of space-specific and choice-specific reward expectation in the human cortex using electroencephalography (EEG). Spatial cueing of attention elicits an enhancement of posterior *N2pc* and *P300* event-related potential components [22,23], as well as strong suppression of alpha-band power over target-contralateral (relative to target-ipsilateral) electrodes following cue onset [24,25]. Similarly, attention also produces directional biases in microsaccades – small, involuntary fixational eye movements – toward attended locations [30,31]. By contrast, decisional processes are characterized by distinctive neural signatures. For example, in target detection paradigms, a decisional bias for the "Yes" response (target detection) correlates robustly with a decrease in pre-stimulus alpha power, indexing a general increase in baseline excitability [32,33]. We tested whether each form of reward expectation – and the associated psycho-physical parameter modulations – reflected neural signatures associated with attention ($d'$), decision-making (criterion), or both. Moreover, we employed key-press responses rather than saccades to avoid confounding saccade preparation signatures with those of spatial attention. The introduction of a stimulus location with fixed reward contingency allowed us to test whether space-specific or choice-specific reward manipulation at the variable reward location affected the allocation of visuospatial attention differentially across the two locations, a feature notably absent in the previous studies.

Our results show that modulating reward expectation across locations versus choices produces dissociable effects on behavior and neural processes. Only space-specific reward modulation elicits changes in sensitivity and engages established neural signatures of spatial attention; these effects are consistent with the definition of a limited, conserved resource operating across visual hemifields, validating an earlier hypothesis [29]. By contrast, choice-specific reward modulations produce local shifts in decisional criteria and elicit none of the neural signatures associated with spatial attention. Surprisingly, and belying our expectations, we observed signatures of higher spatial attentional allocation, in the form of higher $d'$ and lower reaction time, toward the fixed reward location, not toward the variable reward location, during choice-specific reward modulation. In other words, differential spatial relevance governed by decisional priority does not necessarily bias spatial attention. Our findings uncover fundamentally dissociable mechanisms that mediate the effect of reward expectation on sensory and decisional selection, with critical implications for understanding how reward, attention, and choice are linked in the human brain.

## Results

### Distinct forms of reward expectation dissociably modulate sensitivity and criterion

Participants ($n = 24$) performed a two-alternative forced-choice task with two types of reward-cueing sessions (Figs 1A and S1A), each comprising 12 blocks of 48 trials ($n = 1,152$ total trials per participant). The task involved orientation change detection, with the location relevant for decision-making indicated by a post hoc response probe (Fig 1A, yellow triangle; details below). The two reward-cueing sessions differed as follows: in the "space-specific reward expectation" session, rewards for correct responses differed between the two sides (visual hemifields), whereas in the "choice-specific reward expectation" session, rewards for correct responses differed across choices, although average expected reward remained the same on both sides (Figs 1A, 1C, 1D, 1F, 1G, S1B, and S1G).

Reward contingencies were fixed on one side ("fixed" or FX) throughout a trial block but were manipulated on the other side ("variable" or VR) within "mini-blocks" (10–16 trials each). In space-specific sessions, correct responses – hits (H) and correct rejections (CR) – were rewarded equally, but across mini-blocks the overall reward for correct responses on the VR side was either higher ("VR > FX") or lower ("VR < FX") than the FX side (Fig 1D). In choice-specific sessions, the two correct response types were rewarded equally on the FX side, but differentially on the VR side across mini-blocks ("H > CR", and "H < CR") (Fig 1G) (see SI Methods section on "*Reward-cueing*" for details). Half of the blocks were "gain" blocks where correct responses were rewarded but incorrect responses were not penalized (Fig 1D and 1G; "gain" blocks), while the other half were "loss" blocks where incorrect responses were penalized but correct responses were not rewarded. The penalty contingencies in loss blocks closely paralleled the reward contingencies for gain blocks in both sessions (S1B and S1G Fig "loss" blocks).

No explicit cue was provided at the beginning of each mini-block indicating that a change in reward contingency had occurred on the VR side; participants had to infer the contingency switch by regularly monitoring their performance on that side. Switching rewards after mini-blocks of variable length encouraged implicit, dynamic reward contingency learning. It also ensured that participants keenly tracked the VR side for changes in overall reward (space-specific) or choice-related reward (choice-specific) sessions, respectively, thereby allowing us to study the effects of these reward expectation manipulations relative to the fixed-reward (FX) baseline. Moreover, the mini-block structure allowed participants to become aware of the contingency switch, adapt their behavioral strategies and then maintain these strategies over the course of the mini-block [27].

Both types of reward expectation sessions involved an orientation change detection task. A typical trial progressed as follows: After fixation, two Gabor gratings were presented – one in each visual hemifield – along with a central reward-cue (Fig 1A). The reward cue color indicated the side of fixed versus variable reward (location-color mapping counterbalanced across participants). Following a variable delay (exponential, 300–700 ms), the screen was blanked briefly (200 ms), and the gratings reappeared (200 ms). Upon reappearance, either grating could have changed in orientation or not, independently of the other (50% probability of change versus no change). In the response epoch, a central probe appeared (Fig 1A). Participants were instructed to report whether they had detected a change in the orientation of the grating on the probed side (Yes/No choices). Participants were instructed regarding the reward or penalty contingencies prior to each session, and the block type (gain versus loss) was cued visually with distinct fixation shapes (circle, cross) in each block (Figs 1A and S1A). Subsequently, behavioral results are first described for the "gain" blocks, and then for the "loss" blocks.

Differential reward expectation across spatial locations produced systematic effects on sensitivity alone. Sensitivity was consistently higher at the location of the higher average reward expectation: $d'$ was higher at the VR side when rewards were higher at VR (VR > FX), and vice versa for the other contingency (VR < FX) (Fig 2A). We quantified this sensitivity modulation by measuring the difference in $d'$ between the two reward expectation contingencies ($\Delta d' = d'_{VR > FX} - d'_{VR < FX}$), separately for each location. We observed decisive evidence for a significantly more positive $d'$ modulation ($\Delta d' > 0$) at the VR side (Fig 2B, y-axis) as compared to the FX side ($\Delta d' < 0$) (Fig 2B, x-axis) ($\Delta d'_{FX} = -0.60 \pm 0.12$, $\Delta d'_{VR} = 0.43 \pm 0.09$, Signed rank $p < 0.001$, $BF_{+0} > 10^3$) for the VR > FX relative to the VR < FX condition. The absolute levels of $d'$ modulation were not statistically significantly different across the two sides (signed rank $p = 0.103$, $BF_{10} = 0.93$). $d'$ was comparable, overall, between the two locations ($\mu$-$d'_{FX} = 0.66 \pm 0.10$, $\mu$-$d'_{VR} = 0.71 \pm 0.08$, $p = 0.549$, $BF_{10} = 0.24$). By contrast, criteria were not significantly modulated by differential reward expectation across locations (Fig 2C). In fact, we observed substantial evidence against a difference in criterion modulation by reward contingency ($\Delta c = c_{VR > FX} - c_{VR < FX}$) across the two locations (Fig 2D) ($\Delta c_{FX} = 0.01 \pm 0.09$, $\Delta c_{VR} = 0.10 \pm 0.08$, $p = 0.511$, $BF_{10} = 0.24$). The absolute levels of criterion modulation were also comparable across the two sides ($p = 0.607$, $BF_{10} = 0.24$). ANOVA analyses confirmed these results, with no main effect of side (FX, VR) or reward contingency (VR > FX, VR < FX) on $d'$ but a significant interaction between the two (Table 1, rows 3 and 4).

Complementing this pattern of results, differential reward expectation for the different choices produced robust modulations of criteria alone. Criteria were higher at the VR side when hits (correct Yes choices) were rewarded less than correct rejections (correct No choices) at this location ("conservative"), and vice versa for the other contingency ("liberal") (Fig 2E). We observed very strong evidence for a significantly more negative criterion modulation ($\Delta c = c_{Lib.} - c_{Cons.}$) at the VR side (Fig 2F, y-axis) as compared to the FX side (Fig 2F, x-axis) for the liberal relative to the conservative condition. ($\Delta c_{FX} = -0.07 \pm 0.05$, $\Delta c_{VR} = -0.46 \pm 0.08$, $p < 0.001$, $BF_{-0} > 10^2$). The absolute levels of criterion modulation also showed very strong evidence of significantly greater modulation at VR than the FX side ($p < 0.001$, $BF_{-0} > 10^3$). Criteria were comparable, overall, between the two locations ($\mu$-$c_{FX} = 0.12 \pm 0.05$, $\mu$-$c_{VR} = 0.12 \pm 0.05$, $p = 0.909$, $BF_{10} = 0.22$). Conversely, with this reward manipulation, we found substantial evidence against a difference in sensitivity modulation ($\Delta d' = d'_{Lib.} - d'_{Cons.}$) across the two locations ($\Delta d'_{FX} = 0.05 \pm 0.09$, $\Delta d'_{VR} = 0.15 \pm 0.10$, $p = 0.391$, $BF_{10} = 0.29$) (Fig 2H) (for ANOVA results, see Table 1, rows 5–6). In this case, the absolute levels of $d'$ modulation were comparable across the two sides ($p = 0.864$,

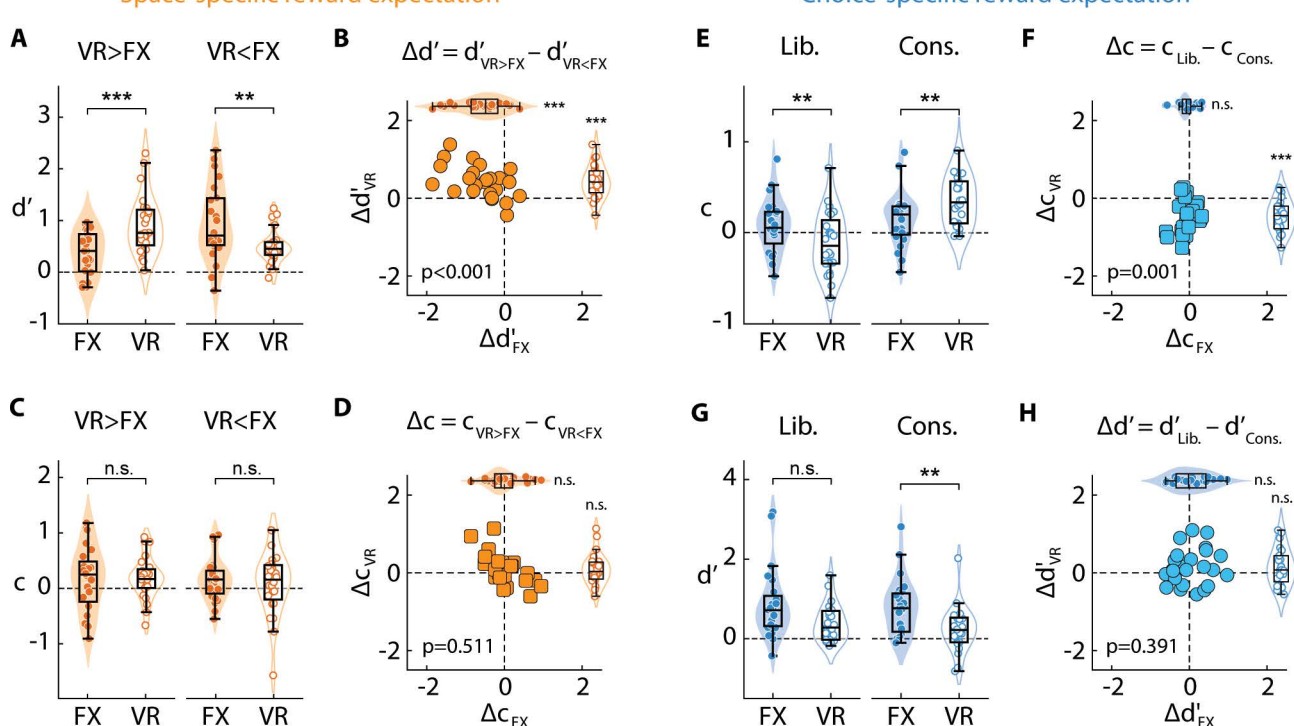

**Fig 2. Space-specific and choice-specific reward expectation independently modulate sensitivity and criterion. A.** Sensitivity ($d'$) for detecting changes at the FX (filled symbols) and VR (open symbols) sides for the two reward contingency conditions ("VR > FX", left pair and "VR < FX", right pair) in the space-specific reward expectation session (gain blocks only). Markers: individual participants ($n = 24$). Box plots limits denote the first and third quartiles; midline: median; whiskers: the minimum and maximum data points not considered outliers. Violin plots: kernel density estimates. Asterisks: statistical significance levels assessed with a Wilcoxon signed rank test (***$p < 0.001$, **$p < 0.01$, *$p < 0.05$, n.s.: not significant). Dashed line: datum ($d' = 0$). **B.** Sensitivity modulation induced by reward contingency change ($\Delta d' = d'_{VR > FX} - d'_{VR < FX}$) at the FX ($x$-axis) and VR ($y$-axis) sides in the space-specific reward expectation session (gain blocks only). $p$-value in the lower left: statistical significance for differences between $\Delta d'_{FX}$ and $\Delta d'_{VR}$. Violin and box plots along the axes: marginals showing $\Delta d'_{FX}$ (filled symbols, horizontal) and $\Delta d'_{VR}$ (open symbols, vertical). Other conventions are the same as in panel **A. C.** Same as in panel **A**, but for criterion ($c$) in the space-specific reward expectation session. Other conventions are the same as in panel **A. D.** Same as in panel **B**, but showing criterion modulation induced by reward contingency change ($\Delta c = c_{VR > FX} - c_{VR < FX}$) in the space-specific reward expectation session. Other conventions are the same as in panel **B. (E–F).** Same as in panels **(C–D)**, respectively, but comparing criteria (**E**) and $c$ modulations (**F**) between the two reward contingencies ("Yes > No" and "Yes < No") in the choice-specific reward expectation session. Other conventions are the same as in panels **C–D. (G–H).** Same as in panels **(A–B)**, respectively, but comparing sensitivities (**G**) and $d'$ modulations (**F**) between the two reward contingencies ("Yes > No" and "Yes < No") in the choice-specific reward expectation session. Other conventions are the same as in panels **A–B**. Data are available at https://doi.org/10.6084/m9.figshare.25966015 [34].

$BF_{10} = 0.21$). Interestingly, we observed a statistically significant main effect of side alone on $d'$ (Table 1, row 5) such that $d'$ for the FX side was numerically greater than that on the VR side (Figs 2G and S1H), possibly because the task was less demanding on the VR side than the FX side in the criterion sessions (see "Discussion").

We repeated these same analyses with the "loss" block trials. Space-specific penalty expectation sessions for the loss blocks were identical to the gain blocks except that, rather than manipulating reward expectation for correct choices, the penalty expectation for incorrect choices (misses and false alarms) was manipulated on the VR side, relative to the FX side (S1B Fig; 50% VR > FX and 50% VR < FX). Similarly, for choice-specific penalty expectation sessions in the loss blocks, the two incorrect choices were penalized equally on the FX side, but differentially on the VR side: penalty for misses (incorrect No responses) was greater than that for false alarms (incorrect Yes responses) ("liberal") on 50% of mini-blocks, and vice versa for the remaining mini-blocks ("conservative") (S1G Fig). A nearly identical pattern of

**Table 1. Reward expectation effects on psychophysical parameters.**

| | | Side | | Reward contingency | | Side × Reward contingency | |
|---|---|---|---|---|---|---|---|
| | | $F_{1,23}$ | $p$ | $F_{1,23}$ | $p$ | $F_{1,23}$ | $p$ |
| **A.** Space-specific rew. exp. | $d'$ | 0.24 | 0.627 | 2.37 | 0.137 | 30.88 | <0.001 |
| | $c$ | 0.46 | 0.504 | 2.28 | 0.145 | 0.27 | 0.607 |
| **B.** Choice-specific rew. exp. | $d'$ | 6.51 | 0.018 | 2.11 | 0.159 | 0.64 | 0.433 |
| | $c$ | 0.00 | 0.985 | 24.86 | <0.001 | 21.26 | <0.001 |

**A.** *F*-statistics and associated *p*-values based on a two-way ANOVA for the main effects of *side* (columns 3–4) and *reward contingencies* (columns 5–6), and their interaction effect (*side × reward contingency*; columns 7–8) on psychophysical parameters – *d'* and *c* (rows 3–4) – in the space-specific reward expectation session. Red text: significant main effect or interaction.

**B.** Same as in Table 1A but for the choice-specific reward expectation session (rows 5–6).

dissociable sensitivity and criterion modulations emerged in the loss blocks also: sensitivity, but not criterion, modulations occurred in space-specific penalty sessions (S1C–S1F Fig) whereas criterion, but not sensitivity, modulations occurred in the choice-specific penalty sessions (S1H–S1K Fig). Moreover, *d'* was highest at the location of highest penalty expectation and criterion was lowest when misses carried a higher penalty expectation than false alarms (S1C and S1J Fig). A detailed comparison of the behavioral effects in the gain and loss blocks (S1 Table) revealed virtually identical modulations of *d'* and c across the corresponding sessions. Moreover, analyses with data pooled across gain and loss blocks also revealed essentially the same patterns of *d'* and c modulations (S1 Table). Because of the nearly identical effects on behavioral parameters in gain and loss blocks, data from the corresponding conditions across both block types were pooled together for all subsequent analyses.

To further assess the relationship between the modulations of *d'* and *c*, we computed the correlation between the *d'* and *c* modulations (Δ*d'* and Δ*c*). These psychophysical parameter modulations were not correlated with each other in either the space-specific reward expectation session (percentage bend correlation coefficient $\rho = 0.10$, $p = 0.237$; permutation test, data pooled across gain and loss blocks; $BF_{10} = 0.13$) or in the choice-specific reward expectation session ($\rho = -0.03$, $p = 0.454$, $BF_{10} = 0.12$). These results further confirmed a strong, systematic dissociation between the modulations of *d'* and *c* by the different reward expectation contingencies.

In summary, manipulating reward expectations across locations or across choices produced a systematic double dissociation in terms of sensitivity and criterion effects, respectively. Differentially modulating reward, or penalty, expectation between locations ("space-specific") produced systematic effects on sensitivity, but not on criterion. Sensitivity was consistently highest at the location at which the highest reward, or penalty, was expected. By contrast, differentially modulating reward, or penalty, expectation between choices ("choice-specific") produced systematic effects on criterion, but not on sensitivity. Criterion was lowest (most liberal) when Yes choices carried a higher expectation of reward, or a lower expectation of penalty, than No responses. Next, we tested if either form of reward expectation recruited neural markers of spatial attention.

## Space-specific, but not choice-specific, reward expectation engages neural markers of spatial attention

Given the robust dissociation between sensitivity and criterion modulations in the space-specific and choice-specific reward expectation sessions, we next tested which, if either, of these modulations engages conventional neural markers of attention. For this we examined established event related potential (ERP) components, such as the posterior *N2pc* and *P300* evoked contralateral to the grating on each side (Fig 3A), and whose amplitudes are reliably modulated by attention [22,23]. In addition, we also quantified the lateralization of alpha-band oscillations, a robust marker of spatial attention [24,25]. Although data from the corresponding conditions across gain and loss blocks were pooled together for these analyses, the results were substantially similar when gain and loss block trials were analyzed separately (3-way ANOVA, see SI Methods section on "*Statistical Tests*").

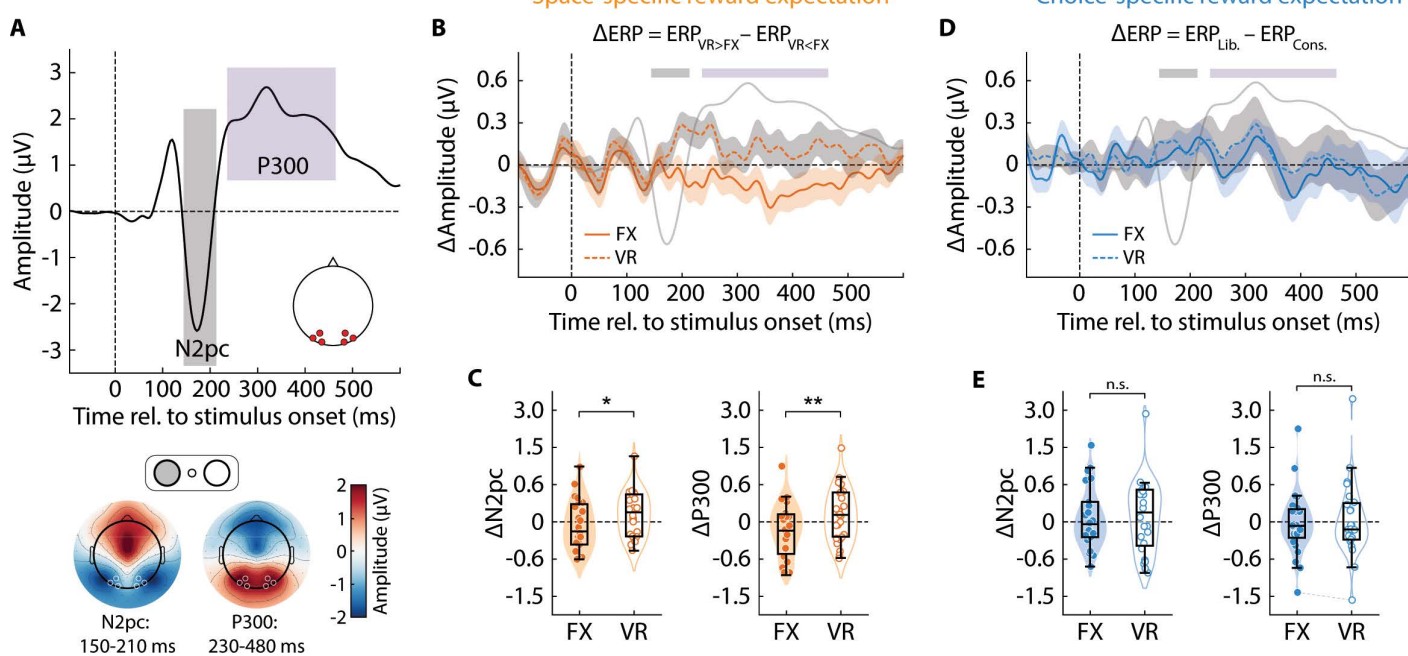

**Fig 3. Space-specific, but not choice-specific, reward expectation modulates attention-related ERPs. A.** (*Top*) ERP waveform from posterior electrodes (see *bottom* panel) in the space-specific reward expectation session (data averaged across $n = 24$ participants and both "gain" and "loss" blocks). *x*-axis: time relative to stimulus onset. Shaded regions: time epochs for quantifying contralateral *N2pc* (grey shading) and contralateral *P300* (purple shading). (*Bottom*) Scalp topography of (*left*) *N2pc* and (*right*) *P300* amplitudes for each electrode. Convention: FX and VR sides are on the left and right visual hemifields, respectively. White circles: occipitoparietal electrodes for ERP quantification. **B.** ERP waveform modulation induced by reward contingency change ($\Delta ERP = ERP_{VR > FX} - ERP_{VR < FX}$) in the space-specific reward expectation session for the FX (solid line) and VR (dashed line) sides. Waveforms were measured from electrodes contralateral to the respective side. Shading on traces (orange and gray): s.e.m. for $\Delta ERP$ traces (FX and VR sides, respectively). Light gray overlaid trace and colored overbars: ERP waveform (not to scale) and time epochs, respectively, for reference, from panel **A. C.** (*Left*) Contralateral *N2pc* component modulation induced by reward contingency change ($\Delta N2pc = N2pc_{VR > FX} - N2pc_{VR < FX}$) for the FX (filled symbols) and VR (open symbols) sides in the space-specific reward expectation session. (*Right*) Same as in the left panel, but for contralateral *P300* component modulation ($\Delta P300 = P300_{VR > FX} - P300_{VR < FX}$). *y*-axis in both panels is plotted in symmetric logarithmic scale. Other conventions are the same as in Fig 2A. Asterisks: statistical significance levels assessed with a Wilcoxon signed rank test (***$p < 0.001$, **$p < 0.01$, *$p < 0.05$, n.s.: not significant). **D.** Same as in panel **B**, but showing ERP waveform modulation induced by reward contingency change ($\Delta ERP = ERP_{Lib.} - ERP_{Cons.}$) in the choice-specific reward expectation session. Other conventions are the same as in panel **B. E.** Same as in panel **C**, but showing contralateral *N2pc* component ($\Delta N2pc = N2pc_{Lib.} - N2pc_{Cons.}$; *left*) and contralateral *P300* component ($\Delta P300 = P300_{Lib.} - P300_{Cons.}$; *right*) modulation in the choice-specific reward expectation session. Other conventions are the same as in panel **C**. Data are available at https://doi.org/10.6084/m9.figshare.25966015 [34].

With space-specific reward expectation, we observed substantial evidence for a significantly more positive modulation of the *N2pc* ($\Delta N2pc = N2pc_{VR > FX} - N2pc_{VR < FX}$) evoked by VR gratings as compared to that evoked by FX gratings (Fig 3B, gray overbar and 3C, *left*; "sensitivity" session) ($\Delta N2pc_{FX} = -0.05 \pm 0.08$ µV, $\Delta N2pc_{VR} = 0.14 \pm 0.08$ µV, $p = 0.018$, $BF_{+0} = 8.22$) for the VR > FX relative to the VR < FX condition – closely paralleling the trends in $d'$ modulation. Yet, the average *N2pc* amplitude was comparable across locations ($\mu N2pc_{FX} = -1.56 \pm 0.35$ µV, $\mu N2pc_{VR} = -1.62 \pm 0.36$ µV, $p = 0.170$, $BF_{10} = 0.64$). The absolute levels of *N2pc* modulation were not statistically significantly different across the two sides ($p = 0.689$, $BF_{10} = 0.24$). By contrast, with choice-specific reward expectation, we observed substantial evidence against any increase in *N2pc* modulation ($\Delta N2pc = N2pc_{Lib.} - N2pc_{Cons.}$) at the VR side compared to the FX side (Fig 3D, gray overbar and 3E, *left*; "criterion" session) ($\Delta N2pc_{FX} = 0.10 \pm 0.11$ µV, $\Delta N2pc_{VR} = 0.13 \pm 0.15$ µV, $p = 0.753$, $BF_{+0} = 0.29$); confirmatory results were obtained with ANOVA analyses of the *N2pc* amplitude (see Table 2, rows 3, 7); the absolute levels of *N2pc* modulation were comparable across the two sides ($p = 0.072$, $BF_{10} = 1.12$). Additionally, the average *N2pc* amplitude was also not different between the two locations ($\mu N2pc_{FX} = -1.34 \pm 0.34$ µV, $\mu N2pc_{VR} = -1.36 \pm 0.34$ µV, $p = 0.424$, $BF_{10} = 0.22$).

Virtually identical trends were obtained with occipitoparietal *P300* amplitudes. We observed strong evidence for a more positive modulation of *P300* amplitudes at the VR side than that at FX side (Fig 3B, purple overbar and 3C, right) ($\Delta P300_{FX} = -0.16 \pm 0.09$ µV, $\Delta P300_{VR} = 0.13 \pm 0.09$ µV, $p = 0.002$, $BF_{+0} = 12.96$), but only with space-specific reward expectation. With choice-specific reward expectation, no significant increase in modulation of *P300* occurred at VR (i.e., the location of criterion modulation) compared to FX (Fig 3D, purple overbar and 3E, right) ($\Delta P300_{FX} = 0.00 \pm 0.13$ µV, $\Delta P300_{VR} = 0.09 \pm 0.18$ µV, $p = 0.549$, $BF_{+0} = 0.60$). The absolute levels of *P300* modulation were not significantly different across the two sides in either the space-specific ($p = 0.932$, $BF_{10} = 0.21$) or the choice-specific ($p = 0.123$, $BF_{10} = 0.44$) session. Again, confirmatory results were obtained with ANOVA analyses of the *P300* amplitude (see Table 2, rows 4, 8). Interestingly, there was substantial evidence that the average *P300* amplitude in the choice-specific session was significantly more positive at FX than at VR ($\mu P300_{FX} = 2.01 \pm 0.31$ µV, $\mu P300_{VR} = 1.91 \pm 0.30$ µV, $p = 0.030$, $BF_{10} = 0.61$) – paralleling the trend observed for average *d′*, but not *c*, in this session.

Another electrophysiological marker – the P2a, a marker for early visual processing and attentional allocation [22] – was not differentially modulated by either type of reward expectation (S2 Fig and S2 Table).

Next, we examined lateralization of occipitoparietal alpha-band oscillations – a robust neural marker of spatial attention (SI Methods). Specifically, we quantified alpha power modulation over posterior (occipitoparietal electrodes) in $a \pm 0.5$ Hz band around each individual's alpha peak frequency (IAF) in a 500 ms window starting 450 ms after stimulus onset (Fig 4A, inset) (SI Methods; see ref. [35]). We observed clear evidence for such lateralized modulation but only when reward expectation differed between spatial locations (Fig 4A and 4B), not between choices (Fig 4D and 4E). Specifically, quantification of alpha power yielded a significantly greater (more negative) suppressive alpha modulation ($\Delta a = a_{VR > FX} - a_{VR < FX}$) contralateral to the VR side than the FX side in the space-specific reward expectation sessions (Fig 4C) ($\Delta a_{FX} = 0.83 \pm 0.35$, $\Delta a_{VR} = -0.79 \pm 0.35$, $p < 0.001$); Bayes Factor provided strong evidence of a difference in suppression ($BF_{-0} = 25.21$). By contrast, in the choice-specific reward expectation sessions, there was no significantly different

**Table 2. Reward expectation effects on neural and motoric metrics.**

| | | Side | | Rew. contingency | | Side × Rew. contingency | |
|---|---|---|---|---|---|---|---|
| | | $F_{1,23}$ | $p$ | $F_{1,23}$ | $p$ | $F_{1,23}$ | $p$ |
| **A. Space-specific rew. exp.** | *N2pc* | 2.51 | 0.126 | 0.36 | 0.552 | 7.42 | 0.012 |
| | *P300* | 6.26 | 0.020 | 0.03 | 0.858 | 8.75 | 0.007 |
| | $\alpha_{post}$ | 0.05 | 0.822 | 0.00 | 0.949 | 10.79 | 0.003 |
| | $\alpha_{pre}$ | 0.01 | 0.906 | 0.03 | 0.867 | 0.02 | 0.903 |
| **B. Choice-specific rew. exp.** | *N2pc* | 0.04 | 0.837 | 0.81 | 0.379 | 0.13 | 0.724 |
| | *P300* | 2.41 | 0.134 | 0.07 | 0.788 | 1.11 | 0.302 |
| | $\alpha_{post}$ | 3.88 | 0.061 | 0.94 | 0.342 | 0.40 | 0.536 |
| | $\alpha_{pre}$ | 3.06 | 0.094 | 0.16 | 0.694 | 4.81 | 0.039 |
| **C. Space-specific rew. exp.** | **MSC** | 0.06 | 0.804 | 0.10 | 0.756 | 10.23 | 0.004 |
| | **RT** | 0.10 | 0.752 | 2.66 | 0.116 | 17.75 | < 0.001 |
| **D. Choice-specific rew. exp.** | **MSC** | 4.89 | 0.037 | 0.20 | 0.657 | 0.12 | 0.731 |
| | **RT** | 6.77 | 0.016 | 0.08 | 0.783 | 0.48 | 0.497 |

**A.** Same as in Table 1A but showing *F*-statistics and *p*-values for a two-way ANOVA on neural metrics – amplitude of *N2pc* potentials, *P300* potentials, post-stimulus alpha-band power and pre-stimulus alpha-band power (rows 3–6, respectively) – in the space-specific reward expectation session. Red text: significant main effect or interaction.

**B.** Same as in **A** but for the choice-specific reward expectation session (rows 7–10).

**C.** Same as in Table 1A but showing *F*-statistics and *p*-values for an ANOVA on motoric metrics – microsaccade rates and reaction times (rows 11–12, respectively) – in the space-specific reward expectation session.

**D.** Same as in **C** but for the choice-specific reward expectation session (rows 13–14).

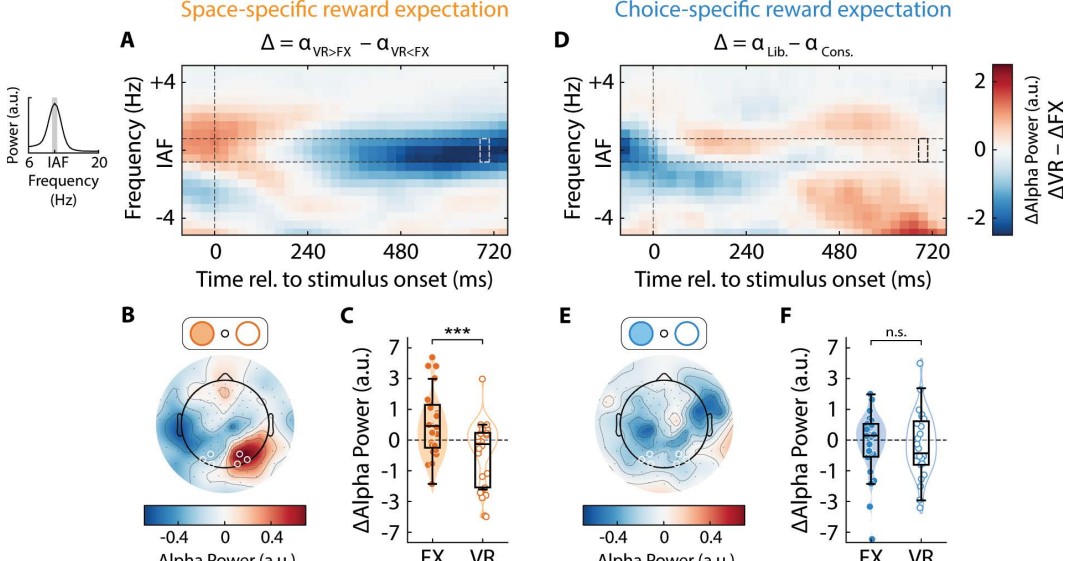

**Fig 4. Space-specific, but not choice-specific, reward expectation produces lateralized alpha power suppression. A.** Time–frequency spectrogram showing the difference in the reward-induced modulation of contralateral alpha power ($\Delta\alpha = \alpha_{VR > FX} - \alpha_{VR < FX}$) between the FX and VR sides ($\Delta\alpha_{FX} - \Delta\alpha_{VR}$) in the space-specific reward expectation session (data averaged across all $n = 24$ participants). x-axis: Time relative to stimulus onset (vertical dashed line); y-axis: frequency relative to the individual alpha frequency (IAF) of each participant (horizontal dashed lines: IAF ± 0.5 Hz). Inset: Alpha-band power spectrum and IAF peak for a representative participant, measured in a 500 ms pre-stimulus time window from occipitoparietal electrodes (see panel **B**). Gray shading: IAF ± 0.5 Hz. **B.** Scalp topography of $\Delta\alpha$ (defined in panel **A**) for each electrode in the space-specific reward expectation session. White circles: occipitoparietal electrodes used for quantifying contralateral IAF alpha power in panel **A**, and for subsequent analyses. Other conventions are the same as bottom panel in Fig 3A. **C.** Reward-induced modulation of contralateral alpha power ($\Delta\alpha$, at IAF ± 0.5 Hz) for the FX (filled) and VR (open) sides in the space-specific reward expectation session. Other conventions are the same as in Fig 2A. Asterisks: statistical significance levels assessed with a Wilcoxon signed rank test (***$p < 0.001$, **$p < 0.01$, *$p < 0.05$, n.s.: not significant). **D.** Same as in panel **A**, but showing the difference in the reward-induced modulation of contralateral alpha power ($\Delta\alpha = \alpha_{Lib.} - \alpha_{Cons.}$) between the FX and VR sides in the choice-specific reward expectation session. Other conventions are the same as in panel **A**. **E.** Same as in panel **B**, but showing the scalp topography of $\Delta\alpha$ for the choice-specific reward expectation session. Other conventions are the same as in panel **B**. **F.** Same as in panel **C**, but showing reward-induced modulation of contralateral alpha power for the choice-specific reward expectation sessions. Other conventions are the same as in panel **C**. Data are available at https://doi.org/10.6084/m9.figshare.25966015 [34].

suppressive alpha modulation across sides ($\Delta\alpha = \alpha_{Lib.} - \alpha_{Cons.}$) ($\Delta\alpha_{FX} = -0.42 \pm 0.41$, $\Delta\alpha_{VR} = -0.17 \pm 0.32$, $p = 0.753$, $BF_{-0} = 0.14$) (Fig 4F), as well as no significant difference in average alpha power between the two sides ($\mu\alpha_{FX} = 12.25 \pm 1.28$, $\mu\alpha_{VR} = 13.22 \pm 1.44$, $p = 0.056$, $BF_{10} = 0.99$). The absolute levels of the post-stimulus alpha suppression modulation were not significantly different across the two sides in either the space-specific ($p = 0.864$, $BF_{10} = 0.21$) or the choice-specific ($p = 0.145$, $BF_{10} = 0.22$) sessions. Confirmatory results were obtained with ANOVA analyses of alpha power lateralization (Table 2, rows 5, 9).

To confirm that the largely null results for the choice-specific reward expectation sessions were not due to baseline differences in signal quality across the session types, we performed two analyses. First, we tested whether the amplitudes of the ERP components (N2pc, P300) or alpha suppression were different, overall, between the session types (SI Methods). Overall, we observed either anecdotal evidence (N2pc: $p = 0.278$, signed rank test; $BF_{10} = 0.44$) or substantial evidence (P300: $p = 0.331$, $BF_{10} = 0.29$; α: $p = 0.587$, $BF_{10} = 0.25$) against this hypothesis. Second, because choice-specific reward expectation induced robust criterion modulations, we investigated EEG markers known to be modulated by response biases; previous studies have identified pre-stimulus alpha power as one such marker [33]. As before, we quantified alpha power around the individual alpha peak (IAF ± 0.5 Hz) in the posterior electrodes, but this time in a pre-stimulus window beginning 500 ms before, until stimulus onset (S3A–S3F Fig). In this case, we observed a pattern of

effects entirely complementary to the patterns reported so far. In the choice-specific reward expectation sessions, modulation of pre-stimulus alpha suppression was substantially more positive at the FX than the VR side ($\Delta a_{FX} = 1.52 \pm 0.94$, $\Delta a_{VR} = -0.66 \pm 1.38$, $p = 0.049$, $BF_{-0} = 3.12$; S3D, S3F Fig). By contrast, in the space-specific reward expectation sessions, we found substantial evidence against an increase in pre-stimulus alpha suppression at FX than at VR locations ($\Delta a_{FX} = -0.36 \pm 2.09$, $\Delta a_{VR} = -0.20 \pm 1.43$, $p = 0.977$, $BF_{-0} = 0.20$; S3A, S3C Fig) (for confirmatory ANOVA results, see Table 2, rows 6, 10). Absolute levels of pre-stimulus alpha suppressive modulation were not statistically significantly different across the two sides in both the space-specific ($p = 0.407$, $BF_{10} = 0.72$) and the choice-specific ($p = 0.092$, $BF_{10} = 0.83$) sessions. Once again, there was no overall difference in pre-stimulus alpha suppression between the two session types ($p = 0.954$, $BF_{10} = 0.31$). Finally, to check if systematic differences appeared in the window leading up to the response, we compared broadband spectral power in the posterior electrodes in a 500 ms window around change onset. Specifically, we compared trials in which participants selected the more biased (high reward or low penalty) choices with those in which they selected the less biased (low reward or high penalty) choices, specifically on the VR side. Although there was a trend in power modulation around change onset in the alpha band, we did not observe any significant spectro-temporal clusters in the window under consideration ($p > 0.05$, cluster-based permutation test, S3G Fig).

In sum, space-specific reward expectation alone engaged established electrophysiological markers of spatial attention. *N2pc* and *P300* amplitudes were higher contralateral to the location of higher reward expectation. Similarly, space-specific reward expectation modulated post-stimulus alpha power such that, alpha power was systematically lower contralateral to the location of higher reward expectation. Neither ERP amplitudes nor post-stimulus alpha power lateralization was affected by choice-specific reward expectation. Nevertheless, an EEG marker for response bias – prestimulus alpha power – was modulated by choice-specific reward expectation alone.

## Space-specific, but not choice-specific, reward expectation modulates motor signatures of spatial attention

Electrophysiological markers of attention over the posterior electrodes typically reflect neural sources linked to visual (input) processing. Next, we tested if either kind of reward expectation would modulate motoric (output) selection signatures associated with spatial attention. When spatial attention is engaged at a location, signatures of selection emerge in at least two motor outputs. First, small, fixational eye movements (microsaccades) are biased toward the attended location during the cue period [30,31]. Second, manual responses are faster for perceptual judgments about the attended target, as compared to the unattended stimulus [36,37].

For the first metric, we quantified oculomotor biases by measuring directional differences in microsaccade rates (MSC) during the cue period (Fig 5A and 5C, thick gray bar; 150–350 ms). In the space-specific reward expectation sessions, microsaccade rates were systematically highest toward the location with the highest reward: in other words, microsaccade rates were higher toward the VR side when rewards were higher at VR (VR > FX; $p = 0.011$, signed rank test; Fig 5A, top), and vice versa for the opposite contingency (VR < FX; $p < 0.001$; Fig 5A, bottom). Microsaccade rates were comparable, overall, between the two locations ($\mu\text{-}MSC_{FX} = 0.09 \pm 0.02$, $\mu\text{-}MSC_{VR} = 0.09 \pm 0.02$, $p = 0.733$, $BF_{10} = 0.22$). Cluster-based permutation test statistics also revealed significantly higher MSC rates occurring during the cue epoch, consistently biased toward the location of the highest reward (Fig 5A, black overline) (VR > FX condition: $p = 0.027$; VR < FX condition: $p = 0.002$; cluster-based permutation test). We observed strong evidence for a significantly more positive modulation of the microsaccade rates ($\Delta MSC = MSC_{VR > FX} - MSC_{VR < FX}$) at the VR side as compared to the FX side ($\Delta MSC_{FX} = -0.04 \pm 0.02$, $\Delta MSC_{VR} = 0.04 \pm 0.01$, $p < 0.001$, $BF_{+0} = 21.02$) (Fig 5B) for the VR > FX than the VR < FX condition; the absolute levels of *MSC* rates modulation were not different across the two sides (signed rank $p = 0.931$, $BF_{10} = 0.23$). ANOVA analyses on microsaccade rates confirmed these findings (see Table 2, row 11).

By contrast, in the choice-specific reward expectation sessions no time window either before or after cue onset showed a statistically significant difference in microsaccade rates between locations or conditions (Fig 5C; $p > 0.05$, cluster-based permutation tests). In addition, we observed substantial evidence against an increase in microsaccade rate modulation

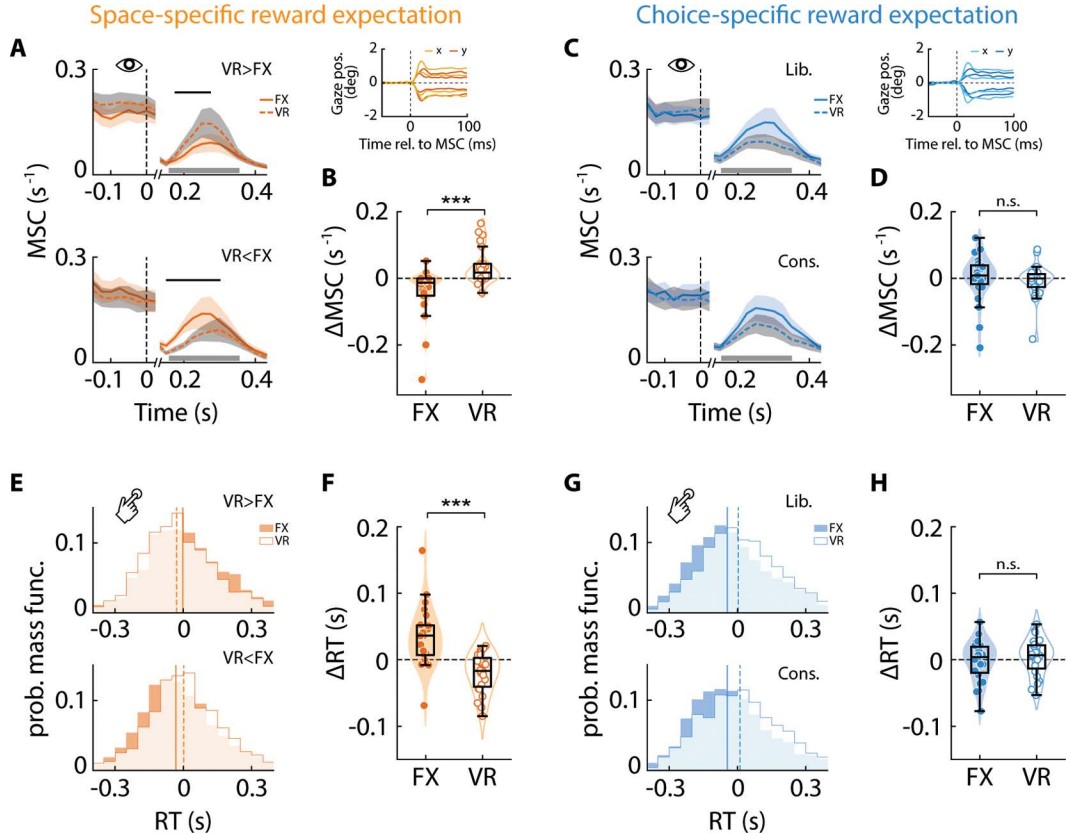

**Fig 5. Space-specific, but not choice-specific, reward expectation biases microsaccades and reaction times spatially. A.** (*Top*) Microsaccade rate (MSC) toward the FX (solid line) and VR (dashed line) sides, respectively, for the "VR > FX" reward contingency in the space-specific reward expectation session (data averaged across $n = 24$ participants). x-axis: time relative to stimulus onset. Shading (orange and gray): s.e.m. for MSC traces (FX and VR sides, respectively). Gray horizontal bar: time epoch for quantifying MSC rates. Black horizontal overline: temporal clusters with significant differences in MSC rates ($p < 0.05$). (*Bottom*) Same as in the top panel, but for the "VR < FX" reward contingency. Inset: Microsaccade traces of horizontal (solid lines) and vertical (dashed lines) gaze positions, for an exemplar participant (SI Methods). **B.** Microsaccade bias induced by reward contingency change ($\Delta MSC = MSC_{VR > FX} - MSC_{VR < FX}$) toward the FX and VR sides in the space-specific reward expectation session. Other conventions are the same as in Fig 2A. Asterisks: statistical significance levels assessed with a Wilcoxon signed rank test (***$p < 0.001$, **$p < 0.01$, *$p < 0.05$, n.s.: not significant). **(C–D).** Same as in panels **(A–B)** and inset, but showing MSC rates, sample traces and bias for the "liberal" and "conservative" reward contingencies in the choice-specific reward expectation sessions. **E.** (*Top*) Distribution of reaction times (RT) for correct responses on the FX (deep shading fill) and VR (unfilled) sides for the "VR > FX" reward contingency in the space-specific reward expectation session (data averaged across $n = 24$ participants). Light shading: region of overlap. Solid and dashed vertical lines: RT medians for the FX and VR sides, respectively. (*Bottom*) Same as in the top panel, but for the "VR < FX" reward contingency. **F.** Same as in panel **B**, but showing reward-induced modulation of reaction times ($\Delta RT = RT_{VR > FX} - RT_{VR < FX}$) in the space-specific reward expectation session. Other conventions are the same as in panel **B**. **(G–H).** Same as in panels **(E–F)**, but showing RTs and their modulation for the "liberal" and "conservative" reward contingency conditions in the choice-specific reward expectation sessions. Data are available at https://doi.org/10.6084/m9.figshare.25966015 [34].

($\Delta MSC = MSC_{Lib.} - MSC_{Cons.}$) toward VR compared to FX ($\Delta MSC_{FX} = 0.00 \pm 0.01$, $\Delta MSC_{VR} = -0.01 \pm 0.01$, $p = 0.224$, $BF_{+0} = 0.17$) (Fig 5D); for ANOVA results, see Table 2 (row 13); absolute levels of MSC rates were not significantly different across the two sides (signed rank $p = 0.198$, $BF_{10} = 0.68$). Nevertheless, there was substantial evidence that overall microsaccade rates were biased toward the FX side in the choice-specific reward expectation sessions ($\mu\text{-}MSC_{FX} = 0.12 \pm 0.02$, $\mu\text{-}MSC_{VR} = 0.08 \pm 0.02$, $p = 0.025$, $BF_{-0} = 3.21$), again consistent with the trend of higher $d'$ toward this location (see "Discussion"). Next, we quantified manual biases by analyzing differences in reaction times for correct responses across the two locations. In the space-specific reward expectation sessions, reaction times were systematically lowest toward the location

with the highest reward: RTs were statistically significantly lower for detecting changes at the VR side when rewards were higher at the VR (VR > FX; $p = 0.049$) (Fig 5E, top), and vice versa for the other contingency (VR < FX; $p = 0.020$) (Fig 5E, bottom). Mimicking the previous results, reaction time modulation ($\Delta RT = RT_{VR > FX} - RT_{VR < FX}$) was significantly more negative at the VR side than at the FX side ($\Delta RT_{FX} = 34.14 \pm 9.24$ ms, $\Delta RT_{VR} = -19.60 \pm 5.97$ ms, $p < 0.001$) (Fig 5F); Bayes factor analysis also provided very strong evidence for this result ($BF_{-0} > 10^2$). Average reaction times were comparable between the two locations ($\mu\text{-}RT_{FX} = 629 \pm 37$ ms, $\mu\text{-}RT_{VR} = 631 \pm 38$ ms, $p = 0.648$, $BF_{10} = 0.23$). Similarly, the absolute levels of reaction time modulation were comparable across the two sides ($p = 0.130$, $BF_{10} = 1.06$). Again, ANOVA analyses of the reaction times produced confirmatory results (Table 2, row 12).

By contrast, in the choice-specific reward expectation sessions, RT modulation ($\Delta RT = RT_{Lib.} - RT_{Cons.}$) was not statistically significantly different between the two locations ($\Delta RT_{FX} = -1.46 \pm 6.28$ ms, $\Delta RT_{VR} = 3.97 \pm 5.67$ ms, $p = 0.587$) (Fig 5H); Bayes factor analysis also provided substantial evidence against a higher modulation (more negative) of RT at VR than at FX ($BF_{-0} = 0.14$). Once again, the absolute levels of reaction time modulation were comparable across the two sides ($p = 0.732$, $BF_{10} = 0.22$). Interestingly, average RTs were significantly lower, overall, at the FX side than at the VR side ($\mu\text{-}RT_{FX} = 642 \pm 25$ ms, $\mu\text{-}RT_{VR} = 684 \pm 26$ ms, $p = 0.021$, $BF_{+0} = 6.49$) (Fig 5G) – a result consistent with the $d'$ trends in these sessions (see "Discussion"); for ANOVA results, see Table 2 (row 14). Additionally, we also tested if choice-specific differences in reward expectation would modulate RTs at the VR side for the more versus less biased choices. In this case, we found that RTs for high reward (low penalty) choices were significantly faster than for low reward (high penalty) choices ($\delta RT = -24 \pm 9$ ms, $p = 0.009$, $BF_{+0} = 8.88$).

In summary, motoric markers of spatial attention were systematically modulated by space-specific, but not choice-specific reward expectation. Reaction times were faster and rates of microsaccades were higher toward the location with higher reward. Nevertheless, manual responses were substantially faster for the more rewarding choices, indicating that participants had planned the corresponding motoric responses well in advance (see "Discussion"). Taken together, the neural and motoric results provide converging evidence that differential expectation for rewards or penalties across locations, but not choices, elicits spatial attentional engagement.

## Only modulations by space-specific reward expectation, but not choice-specific reward expectation, reflect a conserved attentional resource

Finally, we tested if modulations of behavioral parameters, neural or motoric metrics by either form of reward expectation reflect the operation of a conserved resource. For this, we plotted the trial-wise evolution of each of these parameters and metrics locked to reward contingency "switch" trials: the last trial within a reward contingency mini-block after which the participant could have realized that the reward contingency had switched, based on performance feedback (see SI Methods, section on '*Visualizing behavioral and neural modulations induced by reward switch*'). Specifically, we asked whether any parameter or metric would vary on the FX side following the switch trial. Because the reward (or penalty) expectation was fixed and identical on the FX side in all sessions, such a variation would show that changing the reward (or penalty) expectation on the VR side affected the parameter or metric also on the opposite (FX) side, thereby demonstrating cross-hemifield coupling in behavioral performance or neural measures. Similar to the preceding analyses, data were pooled across gain and loss blocks for this and subsequent analyses.

First, we plotted psychophysical parameters ($d'$ and $c$) locked to switch trials. Only $d'$ modulation in the space-specific reward expectation session was consistent with a spatially conserved resource: a decrease (increase) in $d'$ on the VR side was accompanied by a corresponding increase (decrease) in the FX side, as reward contingencies switched (Fig 6A, top and middle rows). By contrast, criterion modulation in the choice-specific reward expectation session revealed no such pattern: even though criteria in the VR side were robustly modulated based on the reward contingency, an analogously strong modulation did not occur on the FX side (Fig 6G, top and middle rows). Plotting the variation in each psychophysical parameter revealed a clear negative relationship between $d'$ values across the two sides, but not in $c$ (compare Fig 6A

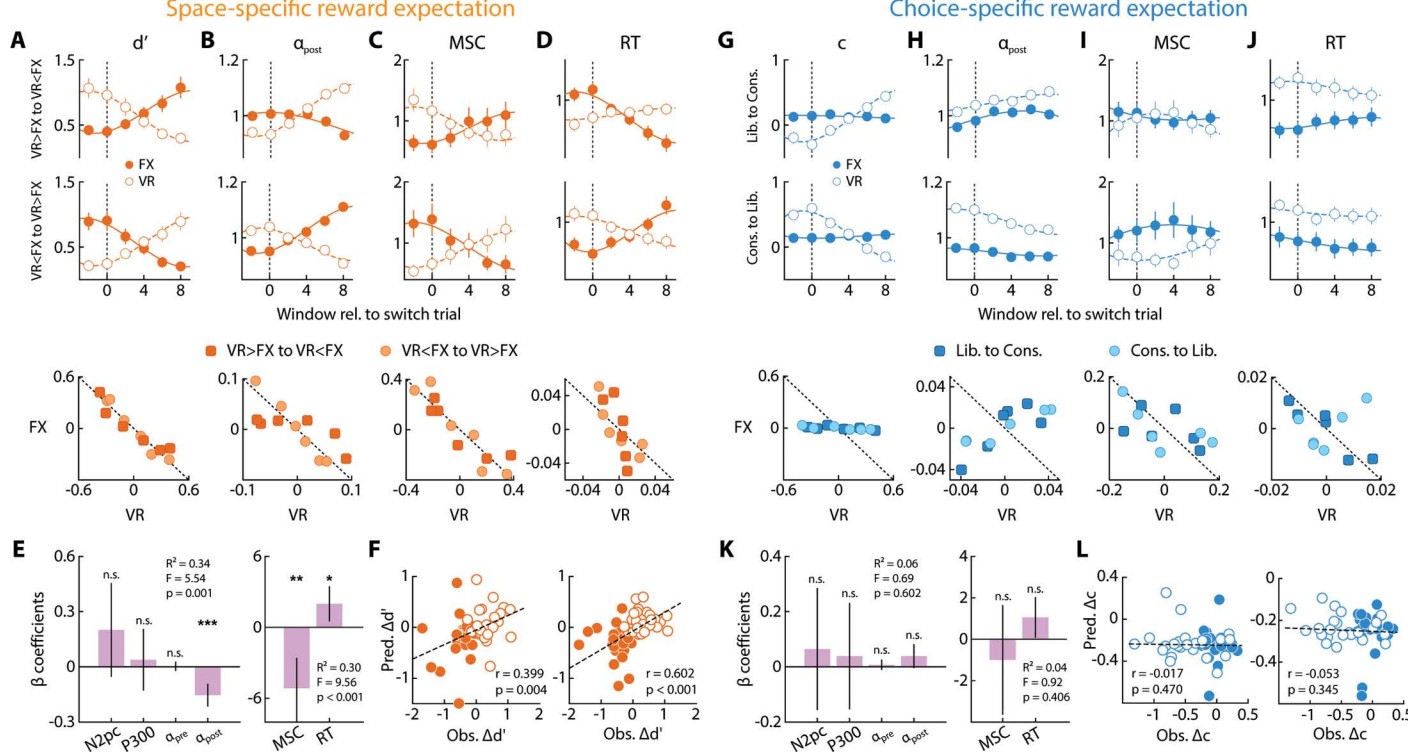

**Fig 6. Space-specific, but not choice-specific, reward expectation engages a conserved attentional resource. A.** (*Top and Middle rows*) Average sensitivity dynamics ($n = 24$ participants) during reward-contingency switch from "VR > FX" to "VR < FX" (*top*) and vice versa (*middle*), measured in sliding windows (width: 9 trials, shift: 2 trials), on the FX (filled circles) and VR (open circles) sides in the space-specific reward expectation session. x-axis: trial count relative to the "switch" trial (SI Methods). Lines: cubic polynomial fits for FX (solid) and VR (dashed) sides. Error bars: s.e.m. (*Bottom row*) $d'$ on the VR side (x-axis) plotted against $d'$ on the FX side (y-axis) in individual windows. Light filled circles and dark filled squares: "VR > FX" to "VR < FX" switch, or vice versa, respectively. Dashed line: $x + y = 0$. **(B–D)**. Same as in panel **A** but showing normalized post-stimulus alpha power suppression ($\alpha_{post}$), microsaccade rates (MSC) and reaction times (RT) dynamics (see text for details), respectively, in the space-specific reward expectation session. Other conventions are the same as in panel **A. E.** Regression coefficient estimates for a model in which reward-induced sensitivity modulation ($\Delta d' = d'_{VR > FX} - d'_{VR < FX}$) was fit with a linear combination of (*left*) neural markers – N2pc amplitude, P300 amplitude, pre-stimulus alpha power ($\alpha_{pre}$), and post-stimulus alpha power ($\alpha_{post}$) modulations – and (*right*) motoric markers – microsaccade rates and reaction time – in the space-specific reward expectation sessions ($n = 24$ participants). Error bars: jackknife s.e.m. Asterisks: statistical significance levels assessed with a permutation test (***$p < 0.001$, **$p < 0.01$, *$p < 0.05$, n.s.: not significant). **F.** Observed reward-induced modulation of sensitivity ($\Delta d'$) (x-axis) plotted against its predicted value (y-axis) with neural (*left*) or motoric (*right*) predictors. Filled and open symbols: $\Delta d' - s$ on the FX and VR sides, respectively. r-values: robust correlations. **G.** Same as in panel **A**, but showing criterion dynamics in the choice-specific reward expectation sessions. **(H–L)**. Same as in panels **B–F** but during the choice-specific reward expectation sessions. Other conventions are the same as in corresponding panels **(A–F)**. Data are available at https://doi.org/10.6084/m9.figshare.25966015 [34].

versus 6G, bottom row). If either of these psychophysical parameters were to fit the definition of a conserved resource across hemifields, their sum would remain constant across sides. Therefore, we de-meaned each psychophysical parameter and fit its variation across the two sides, respectively, to the line $x + y = 0$. This analysis revealed a significantly higher coefficient of determination ($R^2$) for the $d'$ fit, as compared to c fit ($p < 0.001$, permutation test; compare Fig 6A versus 6G, bottom row; fits to the dashed line).

To quantify this effect, we computed sensitivity and criterion modulations ($\Delta d'_{FX}$ and $\Delta c_{FX}$) – the difference in $d'_{FX}$ or $c_{FX}$ values between the two reward contingencies in the space- and choice-specific reward expectation sessions, respectively. $\Delta d'_{FX}$ was significantly negative ($\Delta d'_{FX} = -0.51 \pm 0.08$, $p < 0.001$, $BF_{-0} > 10^4$), indicating that modulating average reward (or penalty) expectation on the VR side affected $d'$ on the FX side in the space-specific reward expectation session.

By contrast, $\Delta c_{FX}$ was not statistically significantly different from zero ($\Delta c_{FX} = -0.03 \pm 0.03$, $p = 0.290$, $BF_{10} = 0.31$), indicating that modulating reward (or penalty) expectation for choices on the VR side did not affect criteria on the FX side in the choice-specific reward expectation session. In other words, sensitivity, but not criterion modulations were indicative of a competitive process acting on a conserved resource across visual hemifields.

We repeated the same analysis with normalized post-stimulus alpha power, reaction times, and microsaccade rates. In each case, we observed a decrease (increase) in $d'$ on the VR side, which was accompanied by a corresponding increase (decrease) on the FX side, as reward contingencies switched (Fig 6B–6D, top and middle rows); a strong competitive effect occurred for all three metrics, but only in the space-specific, not the choice-specific, reward expectation sessions (Fig 6H–6J, top and middle rows). Each of these neural and motoric metrics also exhibited evidence for a robust modulation on the FX side based on the reward contingency on the VR side ($\Delta a_{postFX} = 0.83 \pm 0.35$, $p = 0.037$, $BF_{10} = 2.05$; $\Delta RT = 34.14 \pm 9.24$, $p < 0.001$, $BF_{10} = 29.97$; $\Delta MSC = -0.04 \pm 0.02$, $p = 0.003$, $BF_{10} = 3.12$) in the space-specific reward expectation session. Each of these markers also exhibited a significantly better fit to the $x + y = 0$ line (higher $R^2$) for the space-specific, as compared to the choice-specific, reward expectation sessions ($p < 0.001$, permutation test; compare corresponding panels in Fig 6B–6D versus 6H–6J, bottom row; fits to the dashed line). Interestingly, pre-stimulus alpha power did not show any such competitive effect in either session type, and the modulation was not statistically significantly different from zero on the FX side ($\Delta a_{preFX} = 1.52 \pm 0.94$, $p = 0.331$, $BF_{10} = 0.67$; S3 Fig) in the choice-specific reward expectation session. In sum, post-stimulus (but not pre-stimulus) alpha power, reaction times, and microsaccade rates were governed by a spatially competitive process across visual hemifields.

Given the clear parallels between psychophysical and neural marker modulations by particular kinds of reward expectation, we sought evidence for a stronger relationship between the two quantities. Specifically, we asked if $d'$ or c modulations could be explained reliably from either the neural or motoric marker modulations in each session. For this, first, for each parameter modulation ($\Delta d'$ or $\Delta c$), we fit distinct multiple linear regression models with normalized neural marker modulations ($\Delta N2pc$, $\Delta P300$, $\Delta a_{pre}$, and $\Delta a_{post}$) or motoric marker modulations ($\Delta MSC$ and $\Delta RT$), as the predictor variables (see SI Methods, section on "*Predicting psychophysical parameters with neural and motoric markers*"). We also measured the contribution of each marker toward explaining the variation in the psychophysical parameters based on the magnitude of their regression coefficients ($\beta_i$). In the space-specific reward expectation session, sensitivity modulation ($\Delta d'$) could be reliably explained by the neural markers ($R^2 = 0.34$, $F(4,43) = 5.54$, $p = 0.001$), with a significantly strong contribution of the post-stimulus alpha power modulation ($\beta = -0.16$, permutation test $p = 0.001$) (Fig 6E, left). Similarly, $\Delta d'$ was reliably explained by the motoric markers ($R^2 = 0.30$, $F(2,45) = 9.56$, $p < 0.001$) with significant contributions of both reaction time ($\beta = -5.15$, $p = 0.003$) and microsaccade rate ($\beta = 1.99$, $p = 0.045$) modulations (Fig 6E, right). These results stood in clear contrast to those in the choice-specific reward expectation session: criterion modulations ($\Delta c$) could not be explained reliably by modulations of either the neural ($R^2 = 0.06$, $F(4,43) = 0.69$, $p = 0.602$) or the motoric ($R^2 = 0.04$, $F(2,45) = 0.92$, $p = 0.406$) markers.

To further understand the robustness of this relationship, we also performed a prediction analysis with a linear model [38] to estimate individual $\Delta d'$ or $\Delta c$ values using modulations of either the normalized neural or motoric attentional markers as the predictors (see SI Methods, section on "*Predicting psychophysical parameters with neural and motoric markers*"). Briefly, the method involves fitting linear regression models (same models as described above), with data from all but one participant to estimate regression coefficients ($\beta'_i$). Then the $\Delta d'$ or $\Delta c$ for the left-out participant was predicted with that individual's neural or motoric marker modulations and the $\beta'_i$-s estimated from the previous fit, which excluded that participant (Fig 6F and 6L). Both attentional neural markers as well as the motoric markers robustly predicted individual sensitivity modulations in the space-specific reward expectation session ($\Delta d'$: neural-prediction: $r = 0.399$, $p = 0.004$, $BF_{10} = 2.19$; motoric-prediction: $r = 0.602$, $p < 0.001$, $BF_{10} = 13.02$) (Fig 6F). By contrast, neither of these markers could predict criterion modulations in the choice-specific reward expectation session ($\Delta c$: neural-prediction: $r = -0.017$, $p = 0.470$, $BF_{10} = 0.16$; motoric-prediction: $r = -0.053$, $p = 0.345$, $BF_{10} = 0.16$) (Fig 6L).

In summary, distinct types of reward expectation variations – space-specific and choice-specific – resulted in dissociable modulations of only sensitivity and only criterion, respectively. When reward expectation was varied across spatial locations ("space-specific"), sensitivity was the highest at the location of highest expected reward or penalty. Similarly, when reward expectation was varied between choices ("choice-specific"), criterion was highest when the affirmative choice (Yes) was less rewarded or penalized than the alternative (No). Conventional neural markers of spatial attention – higher posterior ERP (e.g., *N2pc*, *P300*) amplitudes and alpha-band power lateralization – as well as motoric markers – faster response times and directionally biased microsaccades – occurred only during space-specific, but not choice-specific, reward expectation. Furthermore, only sensitivity, and its associated neural and motoric markers, were modulated competitively across hemifields, suggesting an underlying spatially conserved resource. Underscoring these results, both neural and motoric markers of attention systematically predicted sensitivity modulation by space-specific reward expectation, but not criterion modulation by choice-specific reward expectation.

## Discussion

Manipulating rewards or penalties provides a natural and powerful method for engaging spatial attention. Yet, attention is not a unitary concept [39,40]. Recent studies have shown that attentional prioritization can occur either by improving the signal-to-noise ratio for the prioritized sensory information or by enhancing the advantage afforded to prioritized information for decision-making [4,41]. These effects can be quantified with signal detection theory with the sensitivity (*d′*) and criterion (*c*) parameters, respectively [11,12,42,43]. Our study shows that distinct forms of reward expectation – space-specific and choice-specific – dissociably modulated sensitivity and criteria, respectively, and identifies their distinct neural correlates in human cortex.

Do these results imply that the two forms of reward expectation engaged distinct components of attention? We show that this is not the case. Although our study is inspired by recent work [11,12], our results indicate that criterion modulations induced by choice-specific reward expectation do not reflect a component of spatial attention. First, even though criterion modulations occurred specifically at the locations at which rewards were manipulated across choices (VR, not FX) these modulations were not consistent with the operation of a spatially conserved attentional resource operating across hemifields [44]. In fact, even as criteria robustly varied on the variable reward side, no systematic criterion changes occurred concomitantly on the fixed reward side (Fig 6G). By contrast, sensitivity modulations induced by space-specific reward expectation clearly indicated a competitive process for a limited attentional resource allocation, operating across visual hemifields (Fig 6A). In addition, we observed that the modulation of both *d′* and c across reward contingencies exhibited trends, albeit not significant, of an inverse relationship across the two sides (Fig 2B and 2D, *p* > 0.05; space specific reward session). Although the trend in *d′* reinforces our findings of a competitive process (Fig 6A, bottom row), the trend in criterion is not readily explained; we speculate that participants marginally adjusted their criteria also concomitantly with *d′* on each side when reward was space-specific, an observation that must be explored in future studies.

Second, participants had to regularly monitor their performance and feedback on the VR side in both the reward sessions; therefore, the VR location always held the most relevant information to guide behavioral adjustments both in the space-specific and choice-specific reward sessions, respectively. Yet, none of the established neural and motoric signatures of selective spatial attention – including ERP amplitude modulation, alpha lateralization, faster reaction times, and microsaccadic biases – were elicited on the VR side by choice-specific reward expectation, nor could these metrics predict reward-induced criterion modulations on that side. By contrast, each of these signatures was robustly elicited by space-specific reward expectation, and these metrics significantly predicted reward-induced *d′* modulations.

We propose that, in the choice-specific reward expectation session, the shift in the criterion reflects not spatial selection for attention but a differential response strategy across the two locations. Specifically, we hypothesize that participants planned a response to the more rewarding (or less penalizing) alternative, depending on the reward (or penalty) contingency throughout each mini-block of trials. Providing strong evidence for such a response bias, reaction times were

systematically faster for the more rewarding (or less penalizing) response in this session. In other words, criterion modulations arose from a response-planning strategy for each choice-specific reward contingency, not from an attentional choice bias operating across visual fields. Future work with drift-diffusion models could help identify mechanisms by which sensitivity increases towards the more rewarded location versus criterion modulations favoring the more rewarded response alternative influence reaction times (e.g., a change in starting point versus rate of evidence accumulation) [45,46]. Nonetheless, these results indicate that while attentional cueing may alter decisional criteria, thereby providing an advantage for task-relevant information in decision-making, not all types of criterion modulations qualify as attentional components.

Offering suggestive, albeit not conclusive, evidence for this hypothesis, lateralized alpha power was significantly modulated by reward – even before stimulus onset – in the choice-specific reward expectation sessions. This finding is in line with earlier studies. For example, trial-by-trial modulation in lateralized, pre-stimulus alpha activity was linked to corresponding changes in reaction times but not detection rates, suggesting a response-related, rather than an attention-srelated, mechanism [25]. In another study, employing a simple detection task, a decrease in pre-stimulus alpha power was suggested to index an increase in baseline excitability [32], reflecting a greater tendency to report a target regardless of whether the target was actually presented or not [33]. Taken together with these studies, our findings suggest that pre-stimulus alpha power modulation may provide a robust indicator of liberal or conservative response biases, and their associated criterion modulations, in detection tasks.

By contrast, lateralization of post-cue alpha-band oscillations is hypothesized to be a robust indicator for a biased deployment of spatial attention [35]. Post-cue alpha power is known to be modulated robustly over electrodes contralateral to the attended location both before [24,25] and after target onset [35,47]. Suppression of contralateral alpha activity has been linked to facilitated information processing at the attended location, whereas enhancement of ipsilateral alpha activity is suggested to mediate inhibition of stimulus processing at the unattended location [35,48]. Such a decrease in alpha power contralateral to the cued location occurs even in the absence of visual stimulation [25]. In our study, post-stimulus (post-cue) alpha power modulations could systematically predict sensitivity modulations induced by space-specific, but not choice-specific, reward expectation, suggesting that the former alone engaged brain mechanisms linked to spatial attention.

Confirming these conclusions, space-specific reward expectation alone produced systematic modulations of attention-linked ERP amplitudes (*N2pc, P300*). Stronger ERPs are elicited by stimuli at attended locations due to increased sensory gain at these locations [22,23,49]. In visual attention tasks, an increase in the amplitude of the posterior *N2pc* component contralateral to the attended location has been shown to reflect both enhanced sensory processing of the attended stimulus and suppression of the unattended stimulus [50,51] – suggesting that this ERP component serves as a common marker for distinct processes underlying selective attention. The *N2pc* has been observed, specifically in reward-related spatial attention tasks. Higher *N2pc* component amplitudes were observed for high-reward targets, irrespective of their physical salience, in a visual search task, suggesting the *N2pc*'s involvement in orienting endogenous attention [52]. Moreover, in endogenously cued covert spatial attention tasks, the amplitude of the *P300*, a late posterior component, was larger for validly cued targets than for invalidly cued targets [53]. The *P300* amplitude has also been linked to task relevance, priors and reward expectancy of the target stimulus [54–56]. In "oddball tasks" the *P300* is hypothesized to be involved in "context updating" [57]: the periodic updating of the internal representation of the environment to build an accurate model for future decisions. In our task, it is possible that frequent switches in reward contingencies induced a higher target-linked *P300* for the high-rewarding location, suggesting a need for greater vigilance at this location to rapidly identify a change in reward conditions and update attentional focus. Interestingly, we did not observe evidence for modulation of the stimulus-evoked *P2*, a marker of early visual processing and attentional selection [22,58], in either session. It is possible that P2 modulations occurred locked to the onset of the change gratings; unfortunately, the occurrence of the response window immediately following the change gratings precluded a careful analysis of this post-change epoch. Further studies are needed to understand the nature of early ERP component (e.g., *P1, N1, P2*) modulations by reward-induced attention [53].

Additionally, motoric markers revealed a strong and selective link between space-specific reward expectation and spatial attention. Reaction time is a strong indicator of spatial attentional selection, as it decreases at attended locations, both with endogenous and exogenous cueing [59–61]. In our study, we found that reaction times were also significantly faster for response alternatives associated with higher incentives. Reward expectation can also bias oculomotor choices (eye movements) towards stimuli associated with high rewards [27,62]; such biases can persist even when the reward association is no longer present. For example, studies on visual search show that participants bias their saccades towards previously learned locations of high reward, even after those locations were no longer associated with high reward [5,16]. Moreover, in a visually-guided saccade task, saccades made to locations associated with higher reward were more accurate. In addition, microsaccades provide a robust marker for the locus of covert attention spatial [30,63–65] (but see [66]). In endogenous attention tasks, microsaccades directed toward cued locations increased significantly, as early as 200–400 ms after cue onset [63,64]. Similarly, in an exogenously cued attention task, cue-biased microsaccades that occurred in an early epoch (<300 ms) were linked to significant improvements in accuracy and reaction times [30]. Yet, in many previous studies the task design renders the location of higher reward the attentionally relevant location as well as the decisional target for responses. This conflates the effect of reward expectation on spatial attention with its effect on motoric choices (e.g., saccades) that reflect the outcome of the decision, rendering investigation of their respective mechanisms challenging [22,23]. We discounted spatial motoric biases in our task by using key-press responses – instead of saccadic responses; furthermore, these keys were mapped orthogonally (top and bottom keys) to the target locations (left and right hemifields). We showed that microsaccades predicted $d'$ modulation selectively, not criterion modulation. These results further confirm that only $d'$ modulations in the space-specific reward expectation session fit the definition of a spatial attentional component.

Confirmatory evidence for a specific association between $d'$ modulation and neural and motoric signatures of attention emerged from a somewhat unexpected observation. $d'$ was not systematically modulated by choice-specific reward expectation, either on the FX side or on the VR side. Yet, the $d'$ on the FX side was consistently higher than that on the VR side in both the contingency conditions (Figs 2G and S1H), even though the average expected reward (or penalty) was identical between the two sides. Concomitantly, neural markers of attention revealed strong trends of prioritization of the FX side: ERP amplitudes were higher, and alpha amplitudes suppressed, contralateral to the FX side in both the contingency conditions (Table 2, rows 8–10). Similarly, reaction times were significantly faster for, and microsaccades more biased toward, the stimulus on the FX side (Fig 5C and 5G).

These results are surprising in the context of past literature, which suggests that attentional deployment should be higher at the location of higher reward uncertainty. For example, a previous study suggested that during conditions of uncertainty, individuals increase attention to encode information relevant to improving their predictions [67]. Attentional deployment, as quantified by the dwell time of gaze fixation, was higher toward locations with higher uncertainty of reward outcomes [68]. On the other hand, other studies suggest that reward expectation trumps reward uncertainty for guiding attention. For example, when presented with multiple reward-cue contingencies, participants' attention was directed toward locations of higher reward expectancy rather than higher reward uncertainty [15]. Moreover, greater attentional capture occurred at locations of higher reward magnitude, irrespective of the reward uncertainty [62]. In either case, we would have expected $d'$, as well as neural and motoric markers of attention, to have been comparable on the VR and FX sides given their identical average reward expectancy. Alternatively, we would have expected attentional markers to be more strongly expressed on the VR side, because reward changes on the VR side needed to be monitored closely, with additional cognitive resources, given the need to frequently adjust criterion on that side. Our results belied both of these expectations.

We explain this surprising trend in our results as follows: because the choice-specific reward contingencies remained fixed within a mini-block of trials, a suitable response strategy favoring specific choices on the VR side sufficed to enhance accuracy on that side. In contrast, no such response strategy could ensure successful outcomes on the FX side; as a

result, participants allocated more attention toward the FX side in the choice-specific reward expectation sessions. These results have important implications. First, they validate our hypothesis that a response bias strategy was the primary source of criterion modulations by choice-specific reward expectation in this task design. Second, they demonstrate that the neural and motoric markers, such as *N2pc* and *P300* ERP amplitudes, post-cue alpha lateralization, as well as reaction times and microsaccade biases, are related to attention and $d'$ modulation rather than to reward expectation per se. In other words, even when average rewards were equal on the FX and VR sides, neural and motoric markers signaled an advantage for the FX side – the location of higher $d'$ – indicating a specific link between these markers and spatial attention. Third, they show that deploying greater cognitive resources for dynamically adjusting decisional strategies at a location may not result in that location being automatically selected as the focus of attention. In our case, even when relative reward expectations were identical between two target locations, attention was deployed differentially due to other factors like reward uncertainty and relative cognitive demands.

Our study motivates the search for the neural bases of distinct components of attention. We show that criterion effects induced by choice-specific reward manipulations do not qualify as a component of spatial attention. As a result, previous discoveries of the neural basis of such criterion effects in the prefrontal cortex [12] may reflect reward-related processing rather than spatial attention mechanisms, per se. Moreover, while criterion effects induced by choice-specific reward expectation failed to correlate with neural modulations in V4 activity [11], a recent study employing spatial probabilistic cueing reported robust variations of V4 activity with criterion changes [29]. Our study reconciles these seemingly contradictory findings, providing evidence consistent with the latter study. Specifically, we propose that criterion effects that reflect spatial choice biases in localization tasks [42,43], rather than reward-induced response biases, qualify as a component of spatial attention. In fact, choice bias effects are uncorrelated with, and present distinct neural underpinnings from, corresponding sensitivity effects in these tasks [29,69,70]. Moreover, previous work suggests that absolute and relative reward may engage, respectively, distinct signal-to-noise versus downstream selection mechanisms in the brain [4]. Future work will identify the neural basis of absolute and relative reward-induced modulation of sensitivity and bias components of attention.

Additionally, previous work has identified distinct neural bases of reward salience and valence processing in the brain. For example, while the posterior cingulate cortex (PCC) showed engagement during the processing of high reward valence alternatives, the orbitofrontal cortex (OFC) – implicated in reward anticipation [71] – and the anterior cingulate cortex (ACC) – implicated in error prediction [72] – were engaged preferentially during the "win" and "lose" incentives, respectively. Our results showed no main effect of reward valence (gain or loss) on the motoric and neural measures. A potential reason could be the relatively low spatial resolution of surface EEG recordings: dissociating brain regions or mechanisms involved in processing different types of reward may require high-spatial resolution neuroimaging techniques such as functional magnetic resonance imaging (fMRI) [73]. Such techniques may help resolve the neural bases of reward- and penalty-induced modulations of distinct components of attention.

## Materials and methods

### Participants and ethics approvals

Twenty-four volunteers (9 females, 1 left-handed, age range: 20–37 years, mean ± s.e.m. = 26.1 ± 0.9 years) with no known history of neurological disorders, and with normal or corrected-to-normal vision participated in the experiment. The experiment was conducted in accordance with the ethical principles outlined in the Declaration of Helsinki (World Medical Association, 2013) for research involving human participants. All participants provided written, informed consent. Experimental procedures were approved by the Institute Human Ethics Committee (IHEC) at the Indian Institute of Science (IISc), Bangalore (IHEC approval no.: 4-20182020).

De-identified individual data has been made available to reproduce the results (see: https://doi.org/10.6084/m9.figshare.25966015 [34]). The sex and gender of the participants were self-reported. Our study does not include

comparative analyses based on the sex or gender of the participants; therefore, this information has not been considered in the design of this study.

## Behavioral data acquisition and analysis

All participants performed two experimental sessions: a "space-specific" and a "choice-specific" reward expectation session. The entire experiment was conducted over three consecutive days. Participants underwent a staircasing and training session on the first day and then the two main experimental sessions on the second and third days (order counterbalanced across participants), with concurrent electroencephalography (EEG), eye tracking and galvanic skin response recordings (GSR). GSR recordings were not analyzed for this study.

**Task design for manipulating reward expectation.** Participants were seated in an isolated, dark room with their head rested on a chin rest, 60 cm in front of a contrast calibrated display monitor (22-inch BenQ Gw2283 LCD). Stimuli for the experiments were programmed with Psychtoolbox (version 3) and MATLAB R2017b (Natick, MA) on a host system running Windows 10, and participants' responses were recorded using an RB-840 response box (Cedrus, California).

Each trial began with a fixation period of 1,000 ms, during which a central fixation marker (white, 0.42° diameter) was displayed at the center of a grey screen (Fig 1A). Following this, two full contrast Gabor gratings were displayed, one in each visual hemifield (diameter: 9°, spatial frequency: 1.6 cycles per degree, eccentricity: center ±7.8° from the fixation dot). Orientations of the two gratings were sampled independently of each other and pseudorandomly across trials from a uniform random distribution (±15° to ±75° of the vertical). Along with the Gabor gratings, a central "reward" cue (two-sided arrowhead, height: 0.5°, width: 1.3°) also appeared. Each half of the arrowhead was colored differently (Fig 1A; red or blue); these colors indicated the reward (or penalty) expectation contingency on the respective side (see SI Methods section on "*Cueing reward (or penalty) expectation*" for details). Following a variable interval (exponentially distributed, range: 300–700 ms, mean: 550 ms) the screen was blanked briefly (200 ms). Upon reappearance (200 ms), either one, both, or none of the gratings had changed in their orientations. Change probability was 50% on each side and changes on each side occurred independently of the other side. A single orientation change angle was tested; this angle was determined in a previous staircase session for each participant based on an approximately 70% accuracy threshold (range: 6–32°, mean ± std = 16° ± 6°) (see SI Methods section on "*Staircasing and training*"). Following grating disappearance, a central response probe (two-sided arrowhead; same dimensions as the reward-cue) was presented on the screen; the filled (yellow) half indicated the hemifield relevant for response. The participant indicated whether the erstwhile grating on the probed side had changed in orientation ("Yes") or not ("No"), by pressing one of two keys (response window: 1,500 ms). The left and right hemifields were equally likely to be probed for response and were sampled in pseudorandom order across trials. Immediately after the response, audio-visual feedback was provided to inform the participant of their accuracy and cumulative score (see SI Methods section on "*Feedback and monetary compensation*" for details). The mapping of the response keys for Yes and No responses was counterbalanced across participants.

Each experimental session was divided into 12 blocks of 48 trials each. In 6/12 blocks ("gain" blocks) participants were rewarded when they responded correctly (hit or correct rejection), but not penalized otherwise. In the remaining 6 blocks ("loss" blocks), interleaved pseudorandomly with the gain blocks participants were penalized when they responded incorrectly (false alarm or miss), but not rewarded otherwise. The nature of the particular block type ("gain" or "loss") was indicated by the shape of the fixation marker (circle or cross; mapping counterbalanced across participants); participants were informed of this mapping during their training sessions.

*Cueing reward (or penalty) expectation.* Reward (or penalty) expectations for the gain (or loss) blocks followed specific, pre-defined contingencies on each hemifield; these are summarized into two 2 × 2 reward contingency tables (Figs 1D, 1G, S1B and S1G) whose entries reflect the monetary value of reward for correct responses or penalty for incorrect responses in Indian currency (INR or Indian rupee). The reward (or penalty) expectation for one hemifield remained unchanged across an entire a block, which we refer to as the "Fixed" (FX) side. The reward (or penalty) expectation for

the other hemifield, which we refer to as the "Variable" (VR) side, alternated ("switched") between two contingencies across mini-blocks of trials; the length of each mini-block was drawn from an exponential distribution (range: 10–16 trials, mean: 12 trials). Participants used the information from their performance feedback (correct/incorrect response) – specifically, an unexpected non-zero reward (or penalty) – in the VR-side probed trials to learn whether a reward (or penalty) contingency switch had occurred. For each block, the identity of the FX and VR sides was cued based on the fill color (red or blue) of each half of a central double-headed arrow (reward-cue); this color mapping was counterbalanced across participants. FX and VR sides were counterbalanced across blocks in a pseudorandom order.

Participants were tested on two kinds of reward (or penalty) expectation sessions. In "space-specific" reward" expectation sessions, reward expectation was differentially modulated across the FX and VR sides. In these sessions, in the gain blocks, correct responses (hits and correct rejections) were rewarded equally on each side, but average reward magnitudes were different between the two sides (Fig 1D). Similarly, in the loss blocks, incorrect responses (false alarms and misses) were penalized equally for each side, but average penalty magnitudes were different between the two sides (S1B Fig). Specifically, in half of the mini-blocks, correct responses were rewarded more, or incorrect responses penalized more, on the VR side than the FX side (Figs 1D and S1B); in the other half, correct responses were rewarded less, or incorrect responses penalized less, on the VR side than the FX side. In the text we refer to these contingencies as "VR > FX" and "VR < FX", respectively.

In "choice-specific" reward expectation sessions, reward expectation was differentially modulated across choices in the VR side. In these sessions, in the gain blocks, correct responses were rewarded equally on the FX side but unequally on the VR side; the average reward expectation was not different between the FX and VR sides (Fig 1G). Specifically, on the VR side, in half of the gain mini-blocks, correct Yes responses (hits) were rewarded higher than correct No responses (correct rejections) ("Hit > CR"), and vice versa in the other half ("Hit < CR"). Similarly, in the loss blocks, incorrect responses were penalized equally on the FX side but unequally on the VR side (S1G Fig); the average penalty expectation was not different between the FX and VR sides. Again, in half of the loss mini-blocks, incorrect Yes responses (false-alarms) were penalized more than incorrect No responses (misses) ("FA > Miss"; in other words, "Yes > No"), and vice versa in the other half ("FA < Miss", or "Yes < No"). In the later sections of the text, we refer to the "Hit > CR" reward and "FA < Miss" penalty contingencies as "liberal", and the converse contingencies ("Hit < CR" reward and "FA > Miss" penalty) as "conservative", respectively, to reflect the type of affirmative (Yes) response bias associated with each.

*Feedback and monetary compensation.* Every trial was followed by audio-visual feedback immediately after the response. Accuracy was signaled with a step-up or step-down tone (frequency: 0.5/2 kHz or vice vera, 500 ms duration) delivered through a wired headset (Nokia WH-108). Each type of tone signaled either a correct or incorrect response, and this mapping was counterbalanced across participants (revealed to participants during the training session). In addition, the reward (or penalty) obtained on each trial was displayed centrally, as written text (font height: 0.7°, width: 3–4°) (Fig 1A). Surrounding this text, and concurrently with it, a circular pie chart (diameter: 4.5°) appeared (300–700 ms; Fig 1A); the proportion of fill (white area) indicated the cumulative reward (or penalty) magnitude accrued until the previous trial over the course each block. Then the filled area changed (increase or decrease) to reflect the cumulative reward (or penalty) including the current trial; the change in area was animated for 1,000 ms and remained on screen for an additional 1,000 ms. The pie chart fill reset to zero at the beginning of each block.

Participants were informed prior to the training sessions (see next) that they would be provided with a baseline remuneration of INR 500, and that their goal was to maximize their rewards and to minimize their penalty in the gain and loss blocks, respectively. If no responses were missed, participants could earn a maximum cumulative reward of +INR 360 in gain blocks or incur a maximum penalty of −INR 360 in loss blocks. Trials in which participants missed responding within the response window (1,500 ms) were penalized (loss: –INR 15) and excluded from subsequent analyses; the "no response" penalty was set to a high monetary value in order to deter participants from abstaining from responding on trials when they were unsure of the answer. As result, the proportions of no response trials were negligibly low during the main

experimental sessions (space-specific reward expectation: 0.38±0.13%, mean±std, choice-specific reward expectation: 0.33±0.15%).

In order to ensure that participants maintained a high level of motivation across all blocks in a session, we adopted the following strategy [56]: participants were informed that at the end of each main experimental session, one gain and one loss block would be chosen randomly. At the end of both sessions, the average of the reward and penalty of these chosen sessions would be added to their baseline remuneration. Past literature suggests that such a reward strategy mitigates the influence of reward or penalty accrued in early session blocks on performance in the later blocks [56,74]. Yet, unbeknownst to them, the intimated reward strategy was not exercised when determining the final remuneration: all participants were uniformly awarded INR 900, based on the university's human ethics guidelines for monetary remuneration for time spent in the experiment, including the staircasing, training and the two main sessions.

**Staircasing and training.** On the first day of testing, each participant received training on the task and also performed a staircase session to identify a participant-specific angle of change. The sequence of these sessions was as follows: First, participants performed a brief familiarization session of 40 trials. The task for this session was identical to the main experimental sessions except that no reward cues were provided; an attention cue was provided instead with timings identical with the reward cue. The attention cue was a two-sided arrowhead with the filled half indicating the hemifield to be attended; the cue indicated the side probed subsequently for response with 100% validity. A constant magnitude of angle changes (±25°) was tested. At the end of each trial, participants received visual (text) feedback regarding their accuracy.

Next, participants performed a staircase session with a task protocol identical to the familiarization session, in multiple blocks of 8 trials each with 4 left and 4 right attention cues, pseudorandomly interleaved. The first staircase block began with an orientation change angle of 0°. In each subsequent block, the change angle was incremented or decremented based on the mean accuracy in the current block, as follows: (i) <62.5%, angle increased by 5°; (ii) >75%, angle decreased by 5°; (iii) >62.5% and <75%, angle increased or decreased by 2° depending on whether the performance in the current block was lower or higher than the previous block. Finally, performance was fit with a sigmoid function and the change angle corresponding to 69% accuracy was chosen for the main sessions. In the staircase sessions, no trial-wise feedback was presented to the participants.

Next, participants were trained in both types of reward expectation (space-specific and choice-specific) sessions. Before each training session, all session-specific reward expectation contingencies (Figs 1D, 1G, S1B, and S1G) were presented and explained to the participants. Then each participant performed several training blocks of 48 trials each, with a task protocol identical to that of the main session (see SI Methods section on *"Staircasing and training"*) with audio-visual feedback. In the first two blocks of each training session, on the inter-trial interval immediately prior to the VR reward (or penalty) contingency switched, both reward contingencies (FX and VR) were displayed in the respective hemifield on the screen for 10 s. In subsequent training blocks, as well as in the main sessions, participants were not informed prior to the VR reward (or penalty) contingency switch and were expected to infer the switch based on the reward feedback; behavioral results (Figs 2, 6A, and 6G) indicate that participants were highly successful with inferring the switch. Performance was monitored in each block. At the end of the fourth training block, and at the end of every subsequent block, participants were asked if they had decided on a working strategy to maximize their reward, or minimize their penalty, in the respective session type. Training was stopped when participants reported that they had decided upon a strategy (4-6 training blocks per session per participant). All training sessions comprised an equal number of gain and loss blocks presented in a pseudorandom order.

**Eye-tracking and data exclusion.** Participants' gaze was monitored monocularly throughout the experiment with an infrared eyetracker (Eyelink 1,000 Plus, SR Research, Canada) at 1,000 Hz sampling rate, and stored for offline analysis. Trials were rejected if gaze position exceeded ±1 dva along the azimuthal axis, in an epoch spanning 50 ms before stimulus onset until change offset. Gaze deviated outside this window for >20% of trials for 4 participants in each

of the two session types (2 overlapping participants across sessions). For these participants, all trials were included in the analyses reported in the main text. We also repeated the analyses after excluding these participants, and observed results closely similar to those reported here. For the remaining sessions with stable fixation (20/24 for each session type), the average trial rejection rate was 5.0% (±1.0% s.e.m.) across the space-specific and choice-specific reward expectation sessions.

**Estimating behavioral metrics and parameters.** *Psychophysical parameter estimation.* To analyze the effect of reward expectation on behavior, responses from each participant were summarized into 2 × 2 stimulus-response contingency tables. The rows indicated the stimulus event types ("Change" or "No change") at the probed location, whereas the columns indicated the choice types ("Yes" or "No") for each stimulus event type (Fig 1B). Thus, this table contained four distinct types of stimulus-response contingencies: two types of correct responses – hits (correct report of change) and correct rejections (correct report of no change) – and two types of incorrect responses – misses (incorrect report of no change, when a change had occurred) and false alarms (incorrect report of change, when no change had occurred). The proportion of hits and misses sum to 1.0, as do the proportions of hits and false alarms.

To estimate the psychophysical parameters – sensitivity ($d'$) and criterion ($c$) – we employed a one-dimensional Signal Detection Theory (SDT) model. $d'$ and c were estimated as follows:

$$d' = \Phi^{-1}(HR) - \Phi^{-1}(FAR) \tag{1}$$

$$c = -0.5[\Phi^{-1}(HR) + \Phi^{-1}(FAR)] \tag{2}$$

where, $\Phi^{-1}$ is the inverse of the standard normal cumulative distribution function (probit function), HR denotes hit rates and FAR denotes false alarm rates. The criterion, as estimated here, is inversely proportional to the bias for Yes responses, such that a lower value of c indicates a higher bias to reporting changes at the probed location.

For each session, the psychophysical parameters were estimated, separately, for each side, reward expectation condition, and block type. A three-way ANOVA was performed to compare the effects of side (FX versus VR), reward expectation (VR<FX versus VR>FX, or liberal versus conservative) and block valence (gain versus loss) conditions on psychometric parameters ($d'$, $c$). These revealed no statistically significant two-way or three-way interaction between block valence and any of the other factors ($p > 0.05$ for all). Therefore, we pooled trials across gain and loss blocks of the respective session type for subsequent analyses.

In addition to analyzing the effect of reward expectation contingency on psychophysical parameters with ANOVA, we quantified the modulation of the parameter across different reward (or penalty) expectation contingencies as $\Delta\phi^k = \phi^k_{c1} - \phi^k_{c2}$; where $\phi$ is the psychophysical parameter, the superscript, $k$, indicates the side (FX, VR) and subscripts $c1$ and $c2$ refer to the two reward expectation contingencies (e.g., VR>FX and VR<FX or liberal and conservative). For example, in the space-specific reward expectation session, $\Delta d'$ was calculated as $d'_{VR>FX} - d'_{VR<FX}$. This modulation provides a summary metric quantifying whether and how the respective parameter changes across the two different reward expectation conditions, separately for each side.

*Reaction time and microsaccade analysis.* Participants responded to each trial by pressing one of two keys (5-key response box), with the index finger and thumb of their dominant hand; the mapping between keys and responses (Yes/No) was counterbalanced across participants. Reaction times (RT) were computed as the time interval between the probe onset and response key press. Average RTs reported correspond to only trials with correct responses (hits or correct rejections). To detect microsaccades, we adapted the algorithm proposed by Engbert and Kliegl [63]. We segmented horizontal and vertical gaze traces into 1 s long epochs, from 200 ms before until 800 ms after stimulus onset. For each epoch, we computed bivariate gaze velocities and estimated the median of these velocities across time. A two-dimensional (elliptical) velocity threshold was then estimated as 5× the standard deviation of the velocities across all time

points. Eye movements for which the bivariate velocities exceeded the velocity threshold, and whose final deviations (following the initial transient) were <1 dva were tagged as microsaccades (Fig 5A and 5C, insets). To avoid tagging multiple overlapping threshold-crossing events as distinct microsaccades, we employed a temporal proximity limit such that when events were separated by less than 12 ms, the second event was discarded. In addition, to exclude noisy outlier values, threshold crossing events with peak velocity magnitude exceeding 5× the standard deviation above the mean peak velocity magnitudes were not tagged as microsaccades. Microsaccade rates were then computed over 100 ms windows (sliding by 25 ms; Fig 5A and 5C); microsaccades toward each stimulus were counted as those with an azimuthal component toward the respective stimulus' side and whose direction fell within ±45° polar angle of the horizontal meridian. Even though binocular gaze tracking is known to limit detection errors using a temporal overlap criterion between the two eyes, monocular tracking also suffices in detection of microsaccades, as shown by the similar linear relation between peak velocity and amplitude of both monocular and binocular microsaccades [75].

Similar to previous analyses, the reaction times and microsaccade rates were sorted into the two reward expectation conditions for each side and then averaged across trials in each condition. In addition, we computed the modulation of reaction times ($\Delta RT$) across reward expectation conditions, separately for the FX and VR sides (Fig 5F and 5H). Moreover, to analyze the effect of choice-specific reward expectation on RTs, we quantified the change in the RT at the VR side across high and low biased choices (average across contingencies) as $\delta RT = RT^{bh} - RT^{bl}$, where the superscripts bh and bl refer to high reward/low penalty and low reward/high penalty choices, respectively. Similarly, we computed the microsaccade rate modulation ($\Delta MSC$) in a post-stimulus period, from 150 ms until 350 ms after stimulus onset (Fig 5B and 5D). For each condition, we also performed a one-dimensional cluster-based permutation test to identify the cluster of timepoints after stimulus onset, if any, where the microsaccade rates toward the two locations deviated significantly (Fig 5A and 5C). The clusters of timepoints thus identified considerably overlapped with the 150–350 ms window we used.

## Electrophysiological data acquisition and analysis

**EEG data acquisition.** The electroencephalogram (EEG) was measured with a Net Station acquisition system (version 5.2.0.2, Electrical Geodesics , USA) on a Mac (OS, version 10.10.5) host system and a 128-channels net (HydroCel Geodesic Sensor Net 130, Electrical Geodesics ). Electrode numbers were mapped to the closest electrode coordinates in the international 10−10 system [76]. Channels were referenced to the Cz electrode, data were digitized at 1,000 Hz, and stored for offline analysis. Between blocks, electrode sponges were hydrated with saline (KCl) solution to maintain impedances <50 kΩ (typically <30 kΩ).

**EEG data analysis.** Preprocessing. EEG data were preprocessed using the FieldTrip toolbox [77] in Matlab (version R2021a). EEG data recorded at a sampling rate of 1,000 Hz were bandpass filtered from 0.5 to 35 Hz with an infinite impulse response (IIR) Butterworth filter (filter order: 4). The data were then downsampled to 250 Hz for further analyses. Artifact removal was performed with Independent Component Analysis (ICA); noisy components reflecting artifacts associated with heart rates, blinks, saccades, head movements were manually identified and removed (mean ± std = 1.5 ± 0.9 components rejected per participant). The resultant data was then epoched into from fixation onset until the feedback offset. Further rejection of noisy channels and epochs was performed with SCADS – a procedure for statistical correction of artifacts in dense array studies (SCADS) [78], followed by visual inspection; additional manual rejection of noisy epochs was necessary for one participant with noisy channels in the choice-specific reward expectation session. Rejected channels were then interpolated as the average of the nearest four channels. Epochs were re-referenced to the average of all channels. The remaining epochs were sorted into the two session-specific reward expectation conditions for each side, for all subsequent analyses. For ERP analyses (see next), the epochs were further trimmed to 900 ms: from 200 ms before to 700 ms after stimulus onset; a common baseline signal was subtracted from all epochs based on the mean of a 200 ms pre-stimulus epoch.

*Event-related potential (ERP) analysis*. Average ERP waveforms were computed by averaging epoched EEG traces separately for each side and reward expectation condition, and for channels contralateral to the respective stimulus presentation hemifield. The *N2pc* and P2a components were computed with the occipitoparietal electrodes (PO3/4, PO7/8, O1/2) [50,52] and frontocentral electrodes (F1/2, F3/4, C1/2, C3/4, FC1/2) [79], respectively; the amplitudes of these components were quantified based on their average (negative and positive, respectively) values in a 150–210 ms time window following stimulus onset. The *P300* component was computed from the occipitoparietal electrodes (PO3/4, PO7/8, O1/2) [22,80]; its amplitude was quantified based on the positive average in a 230–480 ms window following stimulus onset. For each component, topographical maps were computed based on ERP amplitude in its respective time window (Figs 3A, bottom and S2A, bottom). As with behavioral metrics, ERP amplitudes were compared between sides and reward expectation conditions both with ANOVAs, as well as with a modulation index across different reward (or penalty) expectation contingencies ($\Delta\phi^k$; see SI Methods section on "*Psychophysical parameter estimation*").

Alpha power analysis. Spectral power was estimated with the Chronux toolbox (version 2.12) using a multi-taper spectral estimation (time × half-bandwidth product = 1; number of tapers = 1). For each trial, the time-frequency spectrogram was estimated in an interval from 1,000 ms before to 1,000 ms after stimulus onset (sliding window size: 500 ms; step size: 25 ms). Analysis windows were padded to achieve a frequency resolution of approximately 0.5 Hz. Alpha-power was computed from the occipitoparietal electrodes (PO3/4, PO7/8, O1/2) (Fig 4B and 4E; topographical map). The individual alpha frequency (IAF) for each participant was determined as the frequency corresponding to the peak power in an extended alpha band (7.5–13 Hz) (IAF range: 7.8–12.2 Hz; mean ± s.e.m. = 10.2 ± 0.2 Hz) in a baseline (500 ms pre-stimulus) window. For plotting spectrograms (Figs 4A, 4D, S3A, and S3D), frequencies were aligned to the IAF as the central frequency, and spectral power in each time window was normalized (divided) by the mean baseline power in a 15–35 Hz band. The normalized alpha power was quantified as its average value either in a window from 450 ms to 950 ms after stimulus onset (post-stimulus alpha power, Fig 4) or in a 500 ms window before onset (prestimulus alpha power, S3A–S3F Fig). Similar to ERP analysis, these power spectra and alpha power data were sorted into the two reward contingency conditions for each side. Alpha lateralization was computed as follows: First we measured an "alpha suppression" metric as the difference in the alpha power between the two reward expectation conditions for each side (FX, VR) from the, respective, contralateral occipitoparietal electrodes. The alpha lateralization was then quantified as the difference in the alpha power suppression metric between the two locations (VR–FX), and the modulation of this difference across reward contingencies was computed separately for each type of reward expectation session (Figs 4C, 4F, S3C, and S3F). To quantify the peri-change alpha suppression in the choice-specific reward expectation session, we followed a procedure similar to that described above, but by computing the spectrogram (time–frequency representation) using 500 ms windows from 500 ms before until 500 ms after change onset (S3G Fig). Here, we defined alpha suppression as the difference in the alpha power – contralateral to the VR side – between the high- and low-biased choices (high reward/low penalty and low reward/high penalty, respectively) averaged across the two contingency conditions (liberal, conservative). We then performed a cluster-based permutation test [81] on this difference to identify time periods and frequency bands with significant alpha suppression.

**Visualizing behavioral and neural modulations induced by reward switch.** To test if modulations of behavioral parameters, neural or motoric metrics reflect the operation of a conserved resource, we visualized the trial-wise evolution of each of these parameters and metrics locked to reward (or penalty) contingency "switch" trials. We defined these switch trials as the first trial within a mini-block of a reward (or penalty) contingency condition during which the participant could have inferred that a reward (or penalty) contingency switch had occurred, based on performance feedback. This typically corresponded to the first trial in a mini-block on which the participant received an unexpected level of non-zero reward for a correct response (or penalty for an incorrect response) on the VR side following the reward (or penalty) switch. For example, in a space-specific reward expectation session (gain block), for a transition from the VR > FX to VR < FX contingency, the switch trial would be the first trial in the VR < FX mini-block in which the VR side was probed

and yielded a low reward rather than a high reward for either correct response (Figs 1D and S1A). Similarly, during choice-specific penalty expectation modulation (loss block), for a transition from the liberal to conservative contingency, the switch trial would be the first trial in the conservative mini-block in which the VR side was probed and yielded high penalty rather than a low penalty for a false-alarm or vice versa for a miss (Figs 1G and S1F). We focused the analysis in Fig 6 around the switch trials to capture a large dynamic range in psychophysical parameters, allowing more reliable estimation of their correlations. To determine the temporal dynamics of attentional shifts, we computed the $d'$ and $c$ parameters (with SDT) within a window from 2 trials before to 14 trials after an inferred contingency switch, averaging over a sliding window of 9 trials (step: 1 trial) to smooth the parameter estimates. To control for inter-participant variability, neural and motoric metrics were normalized for each participant by dividing by the average value over each session of the respective metric for that participant. The average psychophysical and neural metrics across all switch trials of a given type (e.g., VR > FX to VR < FX, or liberal to conservative) were computed for each participant and then averaged across all participants, separately, for each session. In the plots shown in Fig 6 (6A–6D, 6G–6J, top and middle rows), the switch trial, i.e., the trial on which the unexpected level of reward (or penalty) neurofeedback occurred, is marked as trial 0 on the $x$-axis. To understand whether the psychophysical (or neural) metrics across the VR and FX sides were consistent with an attentional resource that was conserved across hemifields, we adopted the following approach: First, we de-meaned the average trace for each metric across windows and then plotted the metrics across the two sides, separately for each session (Fig 6A–6D and 6G–6J, bottom row). Then we fit these curves with the $x + y = 0$ line. We estimated the quality of these fits with the coefficient of determination ($R^2$) and compared these $R^2$ values for the corresponding metrics across the space-specific and choice-specific reward expectation sessions with a random permutation test (see *Statistical Testing*).

**Predicting psychophysical parameters with neural and motoric markers.** To estimate how robustly neural and motoric attentional markers could predict modulations in sensitivity and criterion, we fit linear least squares regression models. The modulation of psychophysical parameters across reward expectation conditions were the response variables ($\Delta d'$ for space-specific, or $\Delta c$ for choice-specific reward expectation sessions, respectively) and either the normalized neural marker modulations (Model I, equation 3) or the motoric marker modulations (Model II, equation 4) were the predictor variables (for details on normalization, see SI Methods section on *Visualizing behavioral and neural modulations induced by reward switch*).

$$\text{Model I}: \Delta\text{d'}(or\ \Delta\text{c}) = \beta_0 + \beta_{N2pc}\Delta\text{N2pc} + \beta_{P300}\Delta\text{P300} + \beta_{pre}\Delta\alpha_{pre} + \beta_{post}\Delta\alpha_{post} + \varepsilon \tag{3}$$

$$\text{Model II}: \Delta\text{d'}(or\ \Delta\text{c}) = \beta'_0 + \beta_{RT}\Delta\text{RT} + \beta_{MSC}\Delta\text{MSC} + \varepsilon \tag{4}$$

where, $\beta_0$ and $\beta'_0$ are intercept terms, $\beta_i$ denote the regression coefficients reflecting the contributions of each predictor variable, $\Delta N2pc$ and $\Delta P300$ represent the modulations of the *N2pc* and *P300* amplitudes, respectively, $\Delta\alpha_{pre}$ and $\Delta\alpha_{post}$ represent the modulations of pre- and post-stimulus alpha power, respectively, and $\Delta RT$ and $\Delta MSC$ represent the modulations of reaction times and microsaccade rates, respectively, across reward expectation contingency conditions, and $\varepsilon$ is the model error (residual). The modulation values for the FX and VR sides for each participant were included in the regression (total of $n = 48 = 2 \times 24$ data points). We tested for multicollinearity – i.e., whether the predictor variables were linearly dependent on each other – and we found no evidence of a strong dependency between any of the predictors in Models I and II in either the space-specific (condition indices < 5 for all) or the choice-specific reward expectation sessions (condition indices < 3 for all). Statistical significance of the $\beta$ coefficients was estimated with a random permutation test (see SI Methods, section on "*Statistical Tests*"). To assess the quality of the model fit, we computed the coefficient of determination ($R^2$) whose statistical significance was assessed with an $F$-test (SI Methods, section on "*Statistical Tests*").

Additionally, we also carried out predictions of participant-specific $\Delta d'$ or $\Delta c$ using a leave-one-out prediction approach [38]. In this case, the models above were trained with data from all participants but one (i.e., $n = 23$ participants in each fold). The regression coefficients thus estimated were used to predict the $d'$ or $c$ modulation for the left-out participant based on their, respective, neural, or motoric marker modulations. This procedure was repeated by leaving out each participant, in turn, yielding predictions of each psychophysical parameter for each participant. Finally, we measured and reported the correlation between the predicted and observed modulations in $d'$ or $c$ across participants, separately for each model, using robust (percentage bend) correlations [82].

## Statistical tests

We employed a three-way ANOVA (2 × 2 × 2) to compare the effects of side (FX, VR), reward expectation conditions and block valence (gain, loss) on the psychophysical parameters ($d'$, $c$), and the neural (ERP amplitudes and alpha power) and motoric metrics (RT, MSC rate). These analyses revealed no statistically significant two-way or three-way interaction between block valence and any of the other factors ($p > 0.05$ for all parameters and metrics). We pooled the data across the two valences and similarly employed a two-way ANOVA (2 × 2) to compare the effects of side and reward expectation conditions on the psychophysical parameters, as well as neural and motoric metrics. For post-hoc multiple comparisons, we used Tukey's HSD (honestly significant difference) test. We employed pairwise non-parametric, Wilcoxon signed rank tests to compare the parameters and metrics between two reward expectation conditions, separately for the two sides. For many of the comparisons reported, we also computed one- and two-tailed Bayes Factors ($BF$), reported as a ratio of the likelihood of the alternative hypothesis to the likelihood of the null hypothesis [83], using JZS priors [84] (JASP 0.17.3). The null hypothesis was an absence of a difference in the reward-driven parameter modulations ($\Delta$) between the locations (FX, VR). We hypothesized that the psychophysical measures – $d'$ or $c$ – would be higher or lower for the location or choice with higher rewards, in the space-specific and choice-specific sessions, respectively. Similarly, we hypothesized that the neural and motoric markers of attention would be higher toward the location of higher sensitivity or lower criterion, in the space-specific and choice-specific sessions, respectively. For metrics that were expected to increase with attention, one-tailed BFs quantified the evidence for an increase in the parameter modulation (e.g., more positive $\Delta d'$, $\Delta N2pc$, $\Delta P300$ and $\Delta MSC$) at the VR than at the FX location, and are denoted as $BF_{+0}$. For metrics that were expected to decrease with attention, one-tailed BFs quantified the evidence for a decrease in parameter modulations (i.e., more negative $\Delta c$, $\Delta \alpha$, and $\Delta RT$) at the VR than at the FX location, and were denoted as $BF_{-0}$. Finally, we did not have an a priori hypothesis regarding the directions of effect of reward on criterion modulations ($\Delta c$) in the space-specific or sensitivity modulation ($\Delta d'$) in the choice-specific sessions, respectively. We also did not have an apriori hypothesis about the direction of the modulation of the average value ($\mu$) of the parameters across locations in either session. Therefore, in these cases, we computed a two-tailed Bayes factor ($BF_{10}$).

To compute the statistical significance of comparisons for the microsaccade rate traces, we employed one-dimensional cluster-based permutation tests [81] to identify temporal windows in which the modulation by reward expectation was significantly different between the two sides. To compare the $R^2$ coefficients fits to the $x + y = 0$ line across the space-specific and choice-specific reward expectation sessions we performed a random permutation test. We shuffled, 1,000 times, the session labels for each parameter pair (FX, VR) – psychophysical, neural, or motoric – across the two sessions. We repeated the model fit to generate a null distribution of the difference between the two $R^2$ coefficients across the two sessions. The one-sided significance $p$-value was calculated as the proportion of the null distribution values that were greater than the difference of the actual $R^2$ coefficients. To quantify how significantly the multiple linear regression models fit the observed modulations ($d'$ or $c$), we computed a test statistic based on an $F$-test, which quantified the goodness of fit of the model relative to modeling the mean alone. To quantify the significance of the beta coefficients from the regression model fit, we performed a permutation test: we shuffled the response variable labels (FX and VR) across participants and repeated the regression model fit for 100,000 iterations to generate a null distribution of beta coefficients of each predictor.

As stated earlier, the one-sided significance *p*-value was calculated as the proportion of the respective null distribution values that were greater than the estimated coefficient. For all correlation analyses reported in the results, unless otherwise mentioned, we employed robust (percentage bend) correlations [82], which correct for marginal outliers.

## Supporting information

**S1 Fig. Effects of space-specific and choice-specific *penalty* expectation on sensitivity and criterion. A.** Same as in Fig 1A (main text), but showing the task schematic of a "loss" block trials. In these blocks participants received a penalty for incorrect responses, but no reward for correct responses. Other conventions are the same as in main Fig 1A. **B.** Same as in Fig 1D (main text) but showing penalty contingencies for the space-specific penalty expectation session ("loss" blocks only). Row and column conventions are the same as in Fig 1D. Numbers within polygons indicate the penalty (in INR, deducted from a baseline remuneration) for the respective, incorrect response type (FA and M). Blanks represent no reward for the correct response types (H and CR). Other conventions are the same as in main Fig 1D. **(C–F).** Same as in Fig 2A–2D (main text), but showing sensitivity (*d′*), criteria (*c*) and their modulations for loss block trials in the space-specific penalty expectation sessions. Other conventions are the same as in Fig 2A–2D, respectively. **G.** Same as in Fig 1G (main text) but showing penalty contingencies for the choice-specific penalty expectation session ("loss" blocks only). Row and column conventions are the same as in Fig 1G. Other conventions are the same as in panel **B** and main Fig 1G. **(H–K).** Same as in Fig 2E–2H (main text), but showing sensitivity (*d′*), criterion (*c*) and their modulations for loss block trials in the choice-specific penalty expectation sessions. Other conventions are the same as in Fig 2E–2H, respectively. Data are available at https://doi.org/10.6084/m9.figshare.25966015 [34].
(PDF)

**S2 Fig. Effects of space-specific and choice-specific reward expectation on the P2 ERP. A.** *Top*: Same as in Fig 3A (main text) but for ERP waveform measured from frontocentral electrodes (see *inset*) in the space-specific reward expectation session (data averaged across all *n* = 24 participants). Shaded regions: time epochs used to quantify anterior P2 event-related potential (grey shading). *Below*: Frontocentral electrodes (white circles) shown on a scalp map from which the ERP was estimated. **(B–C).** Same as in Fig 3B and 3C (main text), but for the contralateral P2 component measured during the space-specific reward expectation session. Other conventions are the same as in main Fig 3B and 3C. **(D–E).** Same as in Fig 3D and 3E (main text), but for the contralateral P2 component measured during the choice-specific reward expectation session. Other conventions are the same as in main Fig 3D–3E. Data are available at https://doi.org/10.6084/m9.figshare.25966015 [34].
(PDF)

**S3 Fig. Choice-specific, but not space-specific, reward expectation produces lateralized pre-stimulus alpha power suppression. A.** Same as in Fig 4A (main text) but showing the reward-induced modulation of contralateral alpha power before stimulus onset (vertical dashed line). Other conventions are the same as in Fig 4A. **(B–C).** Same as in Fig 4B and 4C (main text) but showing pre-stimulus Δ*a* distribution over electrodes, and their mean values in the posterior electrodes (see SI Methods), in the space-specific reward expectation session. Other conventions are the same as in Fig 4B and 4C. **D.** Same as in panel **A**, but for the choice-specific reward expectation session. Other conventions are the same as in panel **A** and main Fig 4A. **(E–F).** Same as in panels **B–C**, but for the choice-specific reward expectation sessions. Other conventions are the same as in panels **B–C** and main Fig 4E and 4F. **G.** Same as in Fig 4A (main text) but showing the modulation of alpha power contralateral to the VR side between high-bias and low-bias choices in the choice-specific reward expectation session. *x*-axis: time relative to change onset. *y*-axis: frequency (5–15 Hz). Other conventions are the same as in main Fig 4A. Data are available at https://doi.org/10.6084/m9.figshare.25966015 [34].
(PDF)

**S1 Table. Effects of reward and penalty expectation on psychophysical parameters. A.** Reward-induced modulation of psychophysical parameters ($\Delta d'$, $\Delta c$) on each side (FX, column 3, and VR, column 4) separately in the gain (reward) and loss (penalty) blocks, as well as with the data pooled across both block types (respective, individual rows), of the space-specific reward or penalty expectation sessions. Column 5: *p*-values for a two-tailed significance test (signed rank). Column 6: two-sided bayes factor. **B.** Same as in **A** but for the choice-specific reward or penalty expectation sessions. (PDF)

**S2 Table. Two-way ANOVA on P2 ERP amplitude. A.** Same as in Table 1A but showing *F*-statistics and *p*-values for an ANOVA on the amplitude of P2 potentials (row 3) – in the space-specific reward expectation session. **B.** Same as in **A** but for the choice-specific reward expectation session (row 4). (PDF)

## Acknowledgments

We thank Raj V. Jain and Priyanka Gupta for their feedback on a preliminary version of this manuscript.

## Author contributions

**Conceptualization:** Devarajan Sridharan.

**Data curation:** Ankita Sengupta.

**Formal analysis:** Ankita Sengupta.

**Funding acquisition:** Devarajan Sridharan.

**Investigation:** Ankita Sengupta.

**Methodology:** Ankita Sengupta, Devarajan Sridharan.

**Project administration:** Devarajan Sridharan.

**Resources:** Devarajan Sridharan.

**Software:** Ankita Sengupta.

**Supervision:** Devarajan Sridharan.

**Validation:** Ankita Sengupta.

**Visualization:** Ankita Sengupta.

**Writing – original draft:** Ankita Sengupta, Devarajan Sridharan.

**Writing – review & editing:** Ankita Sengupta, Devarajan Sridharan.

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
