## [Editor Report · Decision Letter 0]

Dear Devarajan, 

Thank you for submitting your manuscript entitled "A double dissociation between neural mechanisms of reward-driven sensory and decisional selection" for consideration as a Research Article by PLOS Biology.

Your manuscript has now been evaluated by the PLOS Biology editorial staff as well as by an academic editor with relevant expertise and I am writing to let you know that we would like to send your submission out for external peer review.

Once your full submission is complete, your paper will undergo a series of checks in preparation for peer review. After your manuscript has passed the checks it will be sent out for review. To provide the metadata for your submission, please Login to Editorial Manager (https://www.editorialmanager.com/pbiology) within two working days, i.e. by Nov 13 2024 11:59PM.

Kind regards,

Christian

Christian Schnell, PhD

Senior Editor

PLOS Biology

cschnell@plos.org

---

## [Decision Letter · Decision Letter 1]

Dear Devarajan,

Thank you for your patience while your manuscript "A double dissociation between neural mechanisms of reward-driven sensory and decisional selection" was peer-reviewed at PLOS Biology. It has now been evaluated by the PLOS Biology editors, an Academic Editor with relevant expertise, and by several independent reviewers. 

In light of the reviews, which you will find at the end of this email, we would like to invite you to revise the work to thoroughly address the reviewers' reports.

As you will see below, the reviewers Reviewer 1 and Reviewer 2 have a lot of positive comments about your study and mainly ask for clarifications, more methodological details, and some additional analyses. Reviewer 3, in contrast, is more critical, questioning the suitability of the task and the conceptual advance. We think that all concerns need to be addressed, with a particular focus on Reviewer 3's concerns about the framing and presentation of your study and the interpretation.

Given the extent of revision needed, we cannot make a decision about publication until we have seen the revised manuscript and your response to the reviewers' comments. Your revised manuscript is likely to be sent for further evaluation by all or a subset of the reviewers.

**IMPORTANT - SUBMITTING YOUR REVISION**

*Re-submission Checklist*

*Published Peer Review*

*PLOS Data Policy*

*Blot and Gel Data Policy*

Sincerely,

Christian

Christian Schnell, PhD

Senior Editor

PLOS Biology

cschnell@plos.org

REVIEWS:

Reviewer #1: This manuscript by Sengupta and Sridharan presents a perceptual and EEG study on the dissociable effects of reward expectation on sensory processing and decision-making. Using the framework of signal detection theory, the authors ask whether space-specific or choice-specific reward expectation affects behavior in different ways consistent with, or in contrast to, spatial attention effects. Using an orientation change detection task, the authors find that: (1) space-specific reward expectation changed d' while choice-specific reward expectation changed c (criterion), (2) electrophysiological markers of attention (N2pc and P300 amplitudes, alpha suppression) were seen in the space but not the choice reward manipulation, (3) motor correlates of spatial attention (microsaccade rates and reaction times) were driven by space but not choice reward expectation, and (4) only space-specific reward expectation resulted in effects consistent with a shared attentional resource.

Overall, I find the paper to be well written with a clear hypothesis and rigorously quantified results. The topic is a relevant one: reward is a prominent modulator of behavior, cognitive processing, and neural activity but it is often difficult to dissociate the different ways that reward can affect neural activity and resulting behavior. The experiment nicely dissociates reward expectation driven by space and choice, using a button press response to avoid attention-related effects of saccade processing, and the results show a generally consistent pattern of effects across behavior, neural activity, and motor correlates. I have some minor comments the authors can address, but think with revision the paper is worthy of publication.

(1) Relationships between delta_c_VR and delta_c_FX.

In Figure 2B and 2D, there is a marked negative relationship between delta_d'_VR and delta_d'_FX, and between delta_c_VR and delta_cFX. However, I don't believe the authors analyze or address this finding. Does this reflect the effect of individual variation and what might it mean? Perhaps this is worth analyzing and addressing.

(2) Conserved resource analyses

The primary analyses to test the idea of a tradeoff in attentional allocation uses post-switch trials, but it's not quite clear how the authors chose the specifics of the approach. Wouldn't the same tradeoff be observable comparing the blockwise tradeoffs (e.g. d' in VR>FX vs VR<FX)? While the dynamics of the shift is interesting, is the point simply to have more data to analyze - and how exactly did the authors choose the number of post-switch trials to include? IT would help to have a little more justification for this particular analysis.

On a related note, it isn't quite clear what data are being analyzed in lines 616-625. Are these also a subset of trials immediately after the switch? The text state that delta_d'_FX = -0.51 and delta_c_FX = -0.03, but these numbers don't match those in the first results section (lines 205-213) where delta_d'_FX = -0.60 and delta_c_FX = 0.01. Am I missing something, and can the authors clarify why they are different? 

Minor comments

(1) lines 130-132: The current sentence is a bit unclear about # trials per session, suggest editing as: "Participants (n=24) performed a two-alternative forced-choice task with two types of reward-cueing sessions (Fig 1A and S1A Fig), each comprising 12 blocks of 48 trials…" 

(2) line 283: The authors use a percentage bend correlation coefficient rather than a standard one to examine the relationship between delta_d' and delta_c across subjects - can the authors explain this motivation, and was there an issue with outliers?

(3) line 452: probably more accurate to say "electrophysiological markers of spatial attention"

(4) line 523-524: In discussing the higher microsaccade rates to the FX side in the choice manipulation, the authors state it is "consistent with the trend of higher sensitivity toward this location". I think it would be better to use "d'" rather than "sensitivity" for consistency with the rest of the results text, and perhaps to refer specifically back to lines 260-263. A similar suggestion applies for line 542.

Reviewer #2: In this manuscript, Sengupta & Sridharan study the effects of manipulating reward expectation on choice behavior and neural and behavioral markers of spatial attention. Reward expectation is either manipulated in a spatial manner, such that perceptual decisions about stimuli at different spatial locations are rewarded differentially, or in a choice-specific manner, such that Yes (detection of a visual orientation change) and No (no detected orientation change) responses are rewarded differentially. Since a manual response is used for reporting the choice, a potential spatial confound due to eye movement planning is avoided. The authors convincingly show that the spatial manipulation of reward expectation selectively affected sensitivity, whereas the choice-based manipulation of reward expectation selectively affected the criterion. Only the spatial manipulation affected neural and behavioral markers of spatial attention.

The study is well-designed and thorough, the manuscript long and detailed, and fairly well written. The results add to our understanding of neural mechanisms that are responsible for modulating behavior based on reward expectation. I don't have any major concerns, only a smaller issue the authors should look into:

At two locations (lines 203 through 206 and lines 481 through 483) the authors make comparisons between the amount of modulation of some parameter due to the reward manipulation across space (FX vs. VR side). These comparisons should be based on the absolute values of the parameter changes, which indicate the amount of modulation, rather than the signed parameter changes, as currently in the manuscript. In the first case, the sensitivity change on the FX side is -0.60, the sensitivity change on the VR side is +0.43. The signs of the changes have to be opposite, if the larger sensitivity is always on the side with the larger reward, but the larger modulation is happening on the FX side (the one with the larger absolute value of change), not the VR side (the one with the more positive value), as currently stated in the manuscript. In the second case, the change in microsaccades on the FX side is 0.04 vs. +0.04 on the VR side. Again, the sign has to be opposite, if more microsaccades are made towards the side with the larger reward, but the amount of modulation (absolute value of change) appears to be the same on both sides, not larger on the VR side (due to the positive rather than negative sign), as currently stated in the manuscript.

Reviewer #3: In this paper, Sengupta and Sridharan report effects of two different reward manipulations in an orientation-change detection task on both the signal detection theory quantities d' and criterion, and on EEG signals reflecting spatial attention. 

The paper presents some carefully taken and well-presented measurements, but unfortunately I cannot get on board with this study as a candidate for publication in PLOS biology because the paradigm simply does not address what the abstract and intro seem to claim to address. The claim is that the paradigm identifies two distinct forms of reward expectation - guiding attention versus decisions - and establishes whether common or distinct mechanisms mediate them. The task has two versions: the "space-specific" one applies a reward-bias toward a left or right-hemifield source of change-detection evidence, while the "choice-specific" one applies a reward-bias across the index finger/thumb used to report a 'yes'/'no' response, respectively. The outcome is quite trivial: in the first task one side of space is identifiable as more valuable and accordingly, d' and EEG signals confirm that attention is directed there; in the second task, one side is not more valuable than the other and accordingly attention is not spatially biased. It is affirming that this is the case but it does not provide the deep mechanistic insights that are implied in the paper. 

To compound this, another strange (with respect to the reward-cued decision framing) and unexplained feature of the paradigm is that the reward cue is not a reward cue - it indicates the side where the value can flip from high to low or from a high-value-yes to low-value-yes mapping unpredictably from one trial to the next, but it does not indicate the flip itself, and therefore does not explicitly provide information on the value of different sides/response options. This information is surmised implicitly by the subject from experience. It is not made clear why the task was set up in this way or what implications it has for the expected ways the supposed two forms of reward expectation will differ in their mechanisms. It leaves me wondering why the paper wasn't introduced as being focused on that particular feature of implicit, dynamic reward contingency learning. I can imagine that in a more specialised journal, a paper that traces the timecourse of adaptation of EEG/d'/saccade signatures of spatial attention in a task where it should versus shouldn't be expected to adapt to switches in reward assignment, with analyses along the lines of the patterns presented in Figure 6, would be interesting and worthwhile. As it stands, the current framing of the paper fails because it does not convince me of any reason why I should ever have expected signatures of biased spatial attention to arise in a situation where there is in fact no difference in expected reward across spatial locations.

Below I provide some more detailed comments that I think will reflect my confusion as I read through the opening. There was nothing in the Introduction to prepare me for the feature of having a fixed and variable-reward side, which then seemed strange and unmotivated. This is interesting in the Choice-specific task because only one of the potential change-detection evidence sources has a criterion-influencing reward manipulation applied to it, which can reverse unpredictably over time. One might wonder, since there is this cognitively-loaded criterion-adjustment task required for only one side, do we tend to pay more attention to that side? While I find this interesting, the authors apparently do not as not even the task feature itself, let alone the implication I offered, appears anywhere in the framing of the paper. I hope the impression is helpful nonetheless.

The abstract equates 'decision-making' to 'criterion' but of course criterion setting is but one of many aspects of decision-making - to be more precise, 'decision-termination' or decision rule-setting would be more appropriate.

Abstract could also be streamlined, as it is quite dense with jargon and terminology for which it is hard to guess the meaning, e.g. "a globally conserved attentional resource." As I read the abstract I could only loosely guess what this might mean. This reflects a problem also with the Intro: it is missing key, basic information about the task manipulation (especially since the task's ability to 'decouple reward expectation's effects on attention from those on decision-making' seems to be the main innovation) and so it is hard to surmise what is meant by the stated findings.

"signatures of biased decision-making, including pre-stimulus alpha power changes" - can you be more specific about what type of change, since alpha power is known to be sensitive to much more than just decision biases.

I think the opening of the paper needs to articulate the central question more precisely, because at present it reads as "are sensory attention and motor selection one and the same mechanism?" This sounds rather like it is getting at the pre-motor theory of attention and the attention/intention debate. What is meant by 'decision-making' is ambiguous - here it seems to be used to refer to only the result, not the process, of decision-making, and whereas most decision neuroscience considers this to involve a process of gathering sensory evidence over time, here the SDT framework is used, which boils down to a decision rule or criterion without engaging with a decision PROCESS that unfolds over time. More specific operational definitions would help here.

Relatedly, much of the literature review and information motivating the current experiment is hard to follow due to lack of basic information about the tasks being discussed. For example, line 62 raises the problem with previous studies without saying what design feature of previous tasks creates this problem. Is it simply that decisions are reported through actions that target objects separated in visual space, so activity that might look like a motor plan toward a target could relate instead to attention toward that target? 

Another example: line 73 provides a hypothesis about what would happen when reward is manipulated across spatial locations but this doesn't mean anything without stating the role of spatial locations in the task - what is contained at different spatial locations: alternative sources of evidence or alternative targets for action? Again, if the task design is the big innovation it should be explained in the opening.

Line 93 recounts a past study finding "criterion was lowest" at a cued location, but it is not fully clear what this means without more information on that task - was it that subjects had to say whether an orientation happened anywhere, or not, and those changes were more probable at a cued location? Or did subjects only examine the cued location and when that had a greater probability of change, they were more inclined to report a change (i.e. the standard criterion shift effect in SDT?)

The reward cueing was rather complicated, with miniblocks within blocks and assignment of fixed and variable sides. Can the motivation be better explained - why not just have the reward cue point to a low and high-value side, randomised from trial to trial? Were miniblocks cued? That is, were the subjects always aware what sort of miniblock they were in? And doesn't this mean that subjects knew what was high and low value before the cue was even presented? 

Figure 1: A: it should be explained what the red and blue arrow colours mean in the caption - do they indicate the FX and the VR, or do they indicate whether the VR is > or < FX. How do subjects know whether yes or no is more valuable in the choice-specific case? C and F: are these showing the two types of miniblock within a block? The caption for D fails to specify that this is for spatial-specific sessions. 

Line 919: what was the penalisation?

---

## [Decision Letter · Decision Letter 2]

Dear Devarajan,

Thank you for your patience while we considered your revised manuscript "A double dissociation between neural mechanisms of reward-driven sensory and decisional selection" for publication as a Research Article at PLOS Biology. This revised version of your manuscript has been evaluated by the PLOS Biology editors, the Academic Editor and two of the original reviewers.

Based on the reviews and on our Academic Editor's assessment of your revision, we are likely to accept this manuscript for publication, provided you satisfactorily address the remaining points raised by the reviewers and the following data and other policy-related requests:

* We would like to suggest a different title to improve its accessibility for our broad audience: 

Expectation of a reward has distinct effects on sensory processing and decision making in the human brain

* Please include the approval/license number of the ethical approval for the experiments.

* Please include information in the Methods section whether the study has been conducted according to the principles expressed in the Declaration of Helsinki.

* DATA POLICY:

Regardless of the method selected, please ensure that you provide the individual numerical values that underlie the summary data displayed in the following figure panels as they are essential for readers to assess your analysis and to reproduce it: 2ABCDEG, 3CE, 4CF, 5BDFH, 6EK, S1CEHJ, S2CE and S3CF.

* CODE POLICY

We expect to receive your revised manuscript within two weeks. 

*Published Peer Review History*

*Press*

Sincerely,

Christian

Christian Schnell, PhD

Senior Editor

cschnell@plos.org

PLOS Biology

Reviewer remarks:

Reviewer #2: Thank you for thoroughly addressing my previous comments, which have been addressed to my full satisfaction.

Reviewer #3: The authors have been very attentive to the concerns raised in review. My own major concern was that the manipulation that the study was really about - the implicit learning of reward contingencies on a side of the screen that was spatially 'more important,' had been entirely missing in the abstract and intro. The authors have now remedied this and from their replies it seems that this indeed had been the intended focus and it just needed to be clarified. I think that this is emphasised a bit more clearly in the responses to my comments than in the paper itself and a small tweak they can make to help this is to use the same direct phrasing in the paper as they use in the replies - that the task was designed so that even though one side in the choice task was not overall more rewarding, it was *more important* to monitor to follow fluctuating reward contingencies, and thus potentially worthy of attention, whose role was a priori unclear. 

Their responses also make it clear why the reward cues were not direct reward cues in the sense of indicating where the reward is higher.

I thank the authors for their clear responses and congratulate them on a fine paper.

---

## [Editor Report · Decision Letter 3]

Dear Devarajan,

Thank you for the submission of your revised Research Article "Reward expectation yields distinct effects on sensory processing and decision making in the human brain" for publication in PLOS Biology. On behalf of my colleagues and the Academic Editor, Thorsten Kahnt, I am pleased to say that we can in principle accept your manuscript for publication, provided you address any remaining formatting and reporting issues. These will be detailed in an email you should receive within 2-3 business days from our colleagues in the journal operations team; no action is required from you until then. Please note that we will not be able to formally accept your manuscript and schedule it for publication until you have completed any requested changes.

PRESS

We frequently collaborate with press offices. If your institution or institutions have a press office, please notify them about your upcoming paper at this point, to enable them to help maximize its impact. If the press office is planning to promote your findings, we would be grateful if they could coordinate with biologypress@plos.org. If you have previously opted in to the early version process, we ask that you notify us immediately of any press plans so that we may opt out on your behalf.

Sincerely, 

Christian

Christian Schnell, PhD

Senior Editor

PLOS Biology

cschnell@plos.org